# Versatile human cardiac tissues engineered with perfusable heart extracellular microenvironment for biomedical applications

Sungjin Min [1,15], Suran Kim[1,2,15], Woo-Sup Sim [3,4,15], Yi Sun Choi [1], Hyebin Joo[1], Jae-Hyun Park[3,4], Su-Jin Lee[5], Hyeok Kim [3,4], Mi Jeong Lee [1], Inhea Jeong [6], Baofang Cui[1], Sung-Hyun Jo [7], Jin-Ju Kim[3,4], Seok Beom Hong[8], Yeon-Jik Choi[9], Kiwon Ban[10], Yun-Gon Kim[7], Jang-Ung Park[6,11,12,13], Hyang-Ae Lee [5], Hun-Jun Park [3,4,14] ✉ & Seung-Woo Cho [1,2,12,13] ✉

Engineered human cardiac tissues have been utilized for various biomedical applications, including drug testing, disease modeling, and regenerative medicine. However, the applications of cardiac tissues derived from human pluripotent stem cells are often limited due to their immaturity and lack of functionality. Therefore, in this study, we establish a perfusable culture system based on in vivo-like heart microenvironments to improve human cardiac tissue fabrication. The integrated culture platform of a microfluidic chip and a three-dimensional heart extracellular matrix enhances human cardiac tissue development and their structural and functional maturation. These tissues are comprised of cardiovascular lineage cells, including cardiomyocytes and cardiac fibroblasts derived from human induced pluripotent stem cells, as well as vascular endothelial cells. The resultant macroscale human cardiac tissues exhibit improved efficacy in drug testing (small molecules with various levels of arrhythmia risk), disease modeling (Long QT Syndrome and cardiac fibrosis), and regenerative therapy (myocardial infarction treatment). Therefore, our culture system can serve as a highly effective tissue-engineering platform to provide human cardiac tissues for versatile biomedical applications.

Three-dimensional (3D) human cardiac tissues have been widely utilized for developmental studies, disease modeling, drug screening, and regenerative medicine studies[1-4]. Human induced pluripotent stem cell (hiPSC)-derived cardiomyocytes (CMs) have been utilized to fabricate cardiac tissues, as they can recapitulate human physiology and cardiac function[5]. Various engineering techniques, such as electrical and mechanical stimulation, have been applied to promote the maturation of human cardiac tissues that consist of hiPSC-CMs, as achieving maturity is crucial for their functional development[6-10]. Additionally, in recent years, the importance of several microenvironmental factors, such as 3D extracellular matrix (ECM)[11-15], non-myocytes[16-19], and dynamic flow[20,21], has been underscored in engineering human cardiac tissues. While 3D hydrogel scaffolds, such as fibrin, collagen type I, and Matrigel, have been utilized to create ECM-mimicking microenvironments for supporting CM differentiation and maturation[3,22,23], these scaffolds often fall short in reproducing the complex composition and structure of native heart ECM[11,24].

Non-myocytes, including endothelial cells (ECs) and cardiac fibroblasts (CFs), which make up over 70% of the cells in the human heart, play pivotal roles in cardiac development and homeostasis[25–27]. As co-culture of CMs with non-myocytes has predominantly been attempted using simple gravity-mediated aggregation[16,19,28], there is a need to explore more effective cell integration techniques and culture environments that are tailored to all cell types. In addition, optimal fluid flow is a key consideration for supplying oxygen and nutrients for stable long-term culture of large-scale cardiac tissues[20,21]. Until now, the majority of 3D cardiac models have been fabricated to be flat or small (approximately 400 μm in diameter) structures[16,29] to prevent central necrosis, which can limit volumetric expansion and functional maturation. Hence, there is a need to establish effective culture platforms that can overcome the limitations in scale, maturity, and functionality of human cardiac tissues fabricated using existing methods, by incorporating microenvironmental factors that closely resemble in vivo conditions within the heart.

Here, we report the development of a culture platform for engineering human cardiac tissues, which improves cardiac maturation and function, by integrating native heart-like cellular and extracellular components and dynamic flow. Mixed populations of hiPSC-CMs and non-myocytes (ECs and CFs) are co-cultured in 3D heart extracellular matrix (HEM) hydrogel prepared from decellularized porcine heart tissue. The formed HEM-impregnated cardiac tissues are maintained in microfluidic chamber chips with medium axial flow. Our culture method, which recapitulates both cellular and extracellular interactions similar to those found in the native heart, enables the formation of human cardiac tissues with enhanced maturity compared to conventional culture matrices, such as collagen, fibrin, and Matrigel. The well-defined dynamic flow through microfluidic chips allows for stable long-term culture of macroscale (1–1.2 mm) human cardiac tissues, which further enhances structural and functional maturation. We successfully validated the versatile applicability of the human microfluidic HEM cardiac tissue engineered in our culture system for various biomedical applications, including cardiotoxicity prediction (using small molecules with different levels of arrhythmia risk), cardiac disease modeling (such as Long QT Syndrome and cardiac fibrosis), and regenerative therapy (for myocardial infarction treatment).

## Results

### Fabrication of human cardiac tissues equipped with heart cellular and extracellular microenvironments

To generate human cardiac tissues, we established a technique for co-culturing three types of cardiovascular lineage cells (CMs, ECs, and CFs) embedded in decellularized heart tissue-derived HEM hydrogel in a microfluidic chamber chip with dynamic medium flow (Fig. 1a). The HEM was prepared through decellularization of porcine heart tissues and characterized in terms of biocompatibility, batch-to-batch variability, and compositional compatibility. Histological staining and quantitative analyzes revealed that most cells were removed (Fig. 1b), while major ECM components were well preserved in HEM (Fig. 1c, d). The solubilized HEM formed a 3D hydrogel through the self-assembly of collagen nanofibrils, as previously described[24] (Supplementary Fig. 1a).

The biocompatibility of the HEM hydrogel was evaluated in vitro and in vivo (Supplementary Fig. 1b–f). The endotoxin level of the HEM hydrogel was $0.269 \pm 0.003$ EU/ml, which is well below the level set by the Food and Drug Administration (FDA) for medical devices[30] (Supplementary Fig. 1b). RAW 264.7 macrophages incubated with the HEM hydrogel secreted negligible levels of the pro-inflammatory cytokine tumor necrosis factor-α (TNF-α) (Supplementary Fig. 1c). The in vivo biocompatibility of the HEM hydrogel was verified in the subcutaneous space of mice. Hematoxylin and eosin (H&E), as well as toluidine blue (TB) staining, showed that the HEM hydrogel caused no accumulation of inflammatory cells (Supplementary Fig. 1d). The recruitment of α-

smooth muscle actin (α-SMA)$^+$ or Collagen type 1$^+$ activated fibroblasts gradually increased at the injected site, and the recruitment of CD11b$^+$ or CD45$^+$ immune cells increased 1-day post-injection but returned to normal levels thereafter (Supplementary Fig. 1e, f). Thus, there was only a mild foreign body reaction to the HEM hydrogel. The M1/M2 macrophage ratio, calculated by iNOS$^+$ (M1)/CD206$^+$ (M2) or CD80$^+$ (M1)/CD163$^+$ (M2), increased up to day 4 after HEM hydrogel injection, but returned to normal levels by day 7 after injection (Supplementary Fig. 1e, f). This indicates that the HEM hydrogel creates regenerative microenvironments for effective wound repair. Overall, the HEM hydrogel was highly biocompatible in vitro and in vivo.

As HEM is a naturally derived matrix, we assessed the variation of HEM from different batches and donor tissues. Proteomic analysis was conducted to compare the ECM composition in HEMs isolated from different sites of the same porcine donor tissue (Porcine A #1, #2, and #3) and HEMs from different donors (Porcine A, B, and C) (Fig. 1e, f, Supplementary Figs. 2, 3, and Supplementary Data 1). HEMs from different batches and donors exhibited similar profiles in terms of overall matrisome proteins, and the proportions, types, and biological roles of matrisome and non-matrisome proteins were also analogous in all HEM samples (Fig. 1e and Supplementary Figs. 2a, 3). In all HEM samples, there was an overlap of 8 or 9 out of the top 10 matrisome proteins (Fig. 1e). The plots of principal component analysis (PCA) and Pearson's correlation coefficients further supported that all HEM samples exhibited similar profiles in terms of total protein expression (Fig. 1f). Overall, we confirmed that different HEM batches have highly similar ECM composition profiles.

Approximately 17% of the total HEM proteins were identified as heart tissue-enriched proteins, showing at least 4-fold enrichment compared to other tissues, which play crucial roles in the development of CMs and muscle cells (Fig. 1g). The ratio of heart tissue-enriched proteins relative to total proteins was similar in all HEM samples obtained from different batches and donors (Supplementary Fig. 2b). The overall proteomic configuration of the HEM was significantly different from that of growth factor reduced Matrigel (GFR-Matrigel), which is a commonly used commercial matrix (Supplementary Fig. 4). The GFR-Matrigel was primarily composed of glycoproteins, such as laminin proteins as previously reported[24,31], and the number of heart tissue-enriched proteins was significantly lower in the GFR-Matrigel than in the HEM (Supplementary Fig. 4b). The numbers and roles of total proteins and major core matrisome proteins were also distinct between GFR-Matrigel and HEM (Supplementary Fig. 4c, d). Given that HEM contains 80–90 proteins that are present in actual heart tissue, it can serve as a suitable scaffold to fabricate functional cardiac tissues by recapitulating the biological processes in the native heart.

We evaluated the capability of HEM hydrogel to support the co-culture of three types of cardiovascular cells (CMs, ECs, and CFs) in the fabrication of human cardiac tissues. Focusing on gaining a balance between simplicity in fabrication and similarity to native heart tissue, we selected three cell types that effectively represent the major components of the heart – CMs, ECs, and CFs. ECs were included owing to their recognized significance in cardiac development and regeneration[32], as well as their frequent utilization in in vitro production of cardiac models[33]. CFs were included because they are recognized for promoting the maturation of CMs and playing a crucial role in regulating CM function through cellular crosstalk[16]. We utilized human umbilical vein endothelial cells (HUVECs) for ECs, and hiPSC-derived CMs and CFs expressing CM-specific (α-ACTININ) and CF-specific (GATA4, COL1, and DDR2) proteins, respectively (Supplementary Fig. 5). To determine the optimal concentration of HEM hydrogel, human cardiac tissues were developed with CMs, ECs, and CFs utilizing three different concentrations of HEM hydrogels (2, 4, and 6 mg/ml).

Owing to the weaker mechanical properties (elastic modulus) of the HEM hydrogel at lower HEM concentrations (Fig. 1h), the HEM

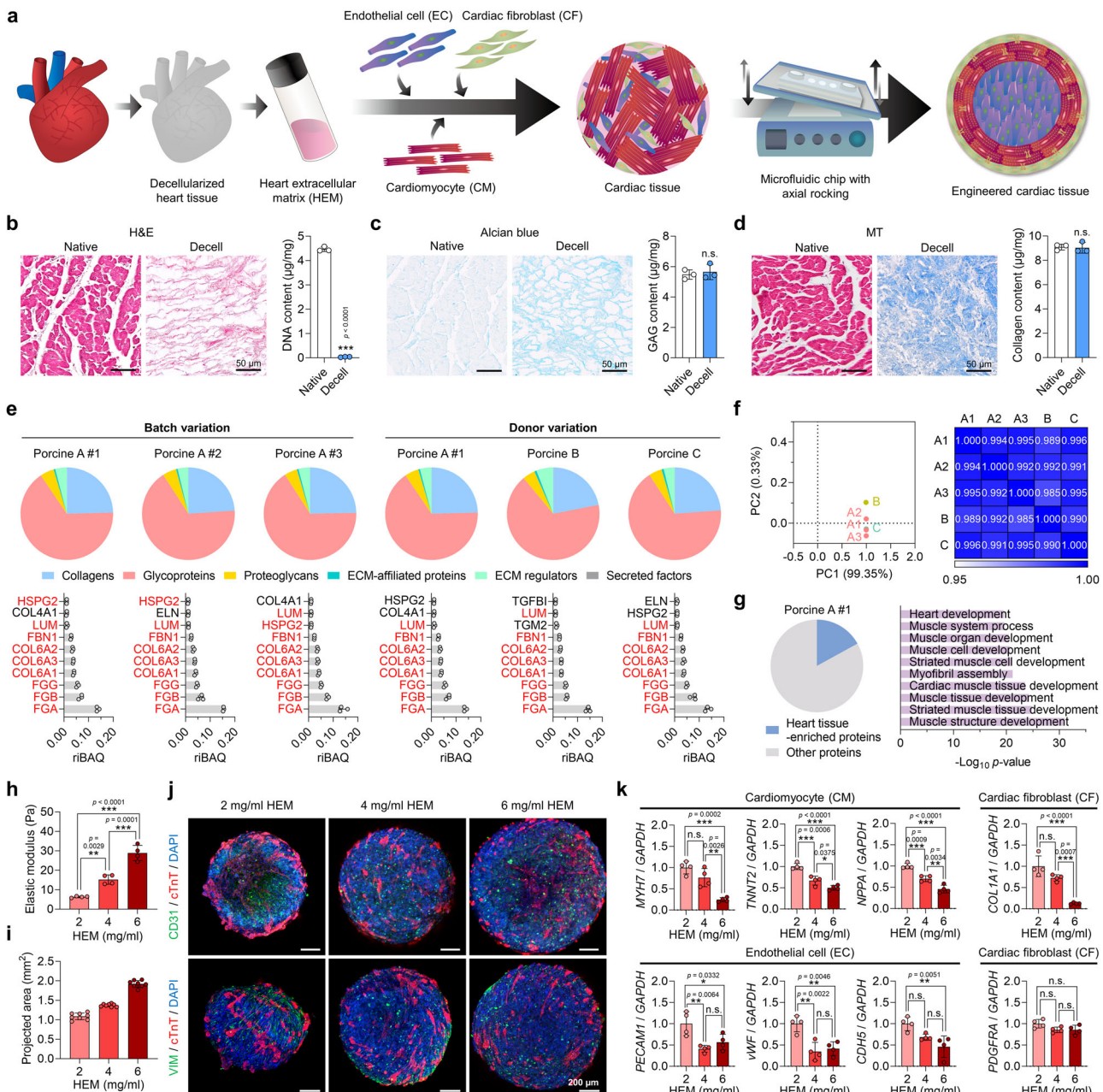

**Fig. 1 | Fabrication of human cardiac tissues with a heart extracellular micro-environment. a** Schematic illustration of human cardiac tissue fabrication using three cell types, heart extracellular matrix (HEM) hydrogel, and a microfluidic chip. **b** Hematoxylin and eosin (H&E) staining and DNA content, **c** Alcian blue staining and glycosaminoglycan (GAG) content, and **d** Masson's trichrome (MT) staining and collagen content of native and decellularized porcine heart tissues (scale bars = 50 μm, $N = 3$, biological replicates, ***$p < 0.001$). **e–g** Proteomic analysis of decellularized porcine tissue-derived extracellular matrix (ECM) samples. **e** ECM composition and the 10 most abundant matrisome proteins in porcine HEM hydrogel samples derived from different batches (A#1, A#2, and A#3) and different donors (A, B, and C) ($N = 3$, biological replicates). Red letters represent proteins that overlapped between different batches or donors. **f** Principal component analysis and Pearson's correlation analysis of total proteins contained in different HEM samples (A#1, A#2, A#3, B, and C). The mean values of three biological replicates were used for analysis. **g** Percentage of heart tissue-enriched proteins (4-fold higher than in other tissues) among total proteins contained in the porcine HEM A#1 sample (left) and Gene Ontology Biological Processes (GOBP) analysis of the

enriched proteins (right). Fisher's exact test with false discovery rate correction was used for data analysis. **h** Elastic moduli of HEM hydrogels produced at concentrations of 2, 4, and 6 mg/ml ($N = 4$, biological replicates, **$p < 0.01$ and ***$p < 0.001$). **i** The projected areas of cardiac tissues fabricated using 2, 4, and 6 mg/ml HEM hydrogel ($N = 8$, biological replicates). **j** Immunofluorescent images of cardiomyocyte (CM; cTnT), endothelial cell (EC; CD31), and cardiac fibroblast (CF; VIM) markers in each group of cardiac tissues (scale bars = 200 μm). **k** Relative mRNA expression levels of CM (*MYH7*, *TNNT2*, and *NPPA*), EC (*PECAM1*, *νWF*, and *CDH5*), and CF (*COL1A1* and *PDGFRA*) markers in each group of cardiac tissues ($N = 4$, biological replicates, *$p < 0.05$, **$p < 0.01$, ***$p < 0.001$, data are representative of two independent experiments. Cardiac tissues prepared using human induced pluripotent stem cell (hiPSC)-derived CMs, CFs, and human umbilical vein endothelial cells (HUVECs) in (**i–k**) were cultured for 7 days and used for analysis. Data are presented as means ± S.D. Statistical significance was determined using unpaired two-sided Student's *t*-tests in (**b–d**) and one-way ANOVA with Tukey's multiple comparisons tests in (**h** and **k**). Non-significant statistical differences are indicated as n.s. ($p > 0.05$).

hydrogel at a concentration of 2 mg/ml formed the most condensed form of cardiac tissues (Fig. 1i). This resulted in the cells being in closer proximity to each other, thus promoting cell-to-cell interactions (Fig. 1j). Accordingly, the expression of three cell type markers, including the CM markers *MYH7*, *TNNT2*, and *NPPA*, the EC markers *PECAM1*, *vWF*, and *CDH5*, and the CF markers *COL1A1* and *PDGFRA*, was the highest in the 2 mg/ml HEM hydrogel (Fig. 1k). Contractile force analysis with electrical pacing indicated that cardiac tissues in 2 mg/ml HEM hydrogel showed the most stable spontaneous beating, and the beating was well regulated in accordance with various conditions of electrical pacing (Supplementary Fig. 6). However, the beating of cardiac tissues was irregular at high concentrations of HEM hydrogel, likely due to a lack of connectivity between the incorporated cells in the HEM hydrogel.

Medium composition was also optimized for co-culture of CMs, ECs, and CFs in HEM hydrogel (Supplementary Fig. 7). A mixture of CM and EC media at a 1:1 (v/v) ratio was the most effective for the growth of CMs and ECs in cardiac tissues. Overall, we determined the optimal HEM hydrogel concentration (2 mg/ml) and culture medium conditions for fabricating human cardiac tissues.

## The superiority of HEM hydrogel for fabricating human cardiac tissues

To establish the superiority of HEM hydrogel for fabricating human cardiac tissues, we compared its culture performance with other fabrication methods, including commercialized hydrogels and U-bottom plates (Fig. 2a). Human cardiac tissues produced in HEM hydrogels were compared with those generated using hydrogel-free suspension cultures in U-bottom plates and 3D cultures in conventional hydrogels, such as collagen gel, fibrin gel, GFR-Matrigel, and embryonic stem cell (ESC)-qualified Matrigel. Although spherical cardiac tissues formed in all test conditions, the two types of Matrigel groups did not induce the formation of compact spherical tissues (Fig. 2b, c) and showed poor cellular connections due to the scattered distribution of incorporated cells (Fig. 2d). Among the tested groups, the HEM hydrogel group formed well-organized cardiac tissues with the most extensive expression of CM and CF markers (Fig. 2d, e). The expression of CM markers (*MYH7*, *TNNT2*, and *NPPA*) and CF markers (*COL1A1* and *PDGFRA*) was similar or higher in the HEM hydrogel group compared to the other hydrogel groups or U-bottom plate group without ECM (Fig. 2f and Supplementary Fig. 8a). The HEM hydrogel groups also expressed EC markers (*PECAM1*, *vWF*, and *CDH5*) at levels comparable to the Matrigel groups. These findings suggest that the HEM hydrogel can provide extracellular microenvironments suitable for CMs and CFs. Overall, HEM hydrogel is more effective for fabricating human cardiac tissues than other methods that utilize conventional hydrogels or plates.

HEM hydrogel was also compared with ECM hydrogels derived from other decellularized tissues, such as stomach, intestine, and muscle (Fig. 2g). Spherical human cardiac tissues formed in all ECM groups, but the HEM hydrogel group exhibited the most compact and condensed constructs (Fig. 2h, i). Tissue organization and cellular connections by CMs were most prominent in the HEM hydrogel group, while CM-positive tissues were insufficient or irregular in the other groups (Fig. 2j). The expression levels of CM and CF markers were generally higher in the HEM hydrogel group compared to hydrogels fabricated from non-heart tissue ECMs (Fig. 2k, l and Supplementary Fig. 8b). These results suggest that there are tissue-specific effects of decellularized tissue-derived ECM hydrogel, likely due to the intrinsic ECM proteins present in each organ[24,31,34]. HEM hydrogel enables the co-development of cardiac and vascular tissues in human cardiac constructs.

HEM hydrogel demonstrated superiority in cardiac tissue construction compared to other hydrogels because of its high level of cell-gel compaction. As shown in the HEM concentration test (Fig. 1h–k),

higher level of cell-gel compaction can be facilitated by the hydrogel with lower elastic modulus, leading to more compact and condensed form of cardiac tissues. This contributes to augmentation of connectivity and interaction between the incorporated cells in cardiac tissues, which is crucial for cardiac differentiation[35]. Therefore, difference in mechanical properties between HEM hydrogels and other hydrogels could affect differentiation and maturation of cardiac tissues (Supplementary Fig. 8c). The larger the elastic modulus of the hydrogel, the lower cell-gel compaction, leading to generation of the cardiac tissue with the larger size as shown in the two types of Matrigel groups (Fig. 2b–d). On the other hand, HEM hydrogel exhibited the best performance for cardiac tissue formation in comparison with other hydrogel groups showing similar levels of elastic modulus and cell-gel compaction, such as fibrin gel, stomach-derived ECM hydrogel, and muscle-derived ECM hydrogel (Fig. 2 and Supplementary Fig. 8c). These data suggest that beneficial effects of HEM hydrogel could be attributed to both high level of cell-gel compaction and heart-specific ECM components.

## Functional improvements of macroscale cardiac tissues achieved through dynamic flow in microfluidic chips

To further improve the functional development of human cardiac tissues, dynamic flow was introduced using a microfluidic chip with axial movement (Fig. 3a and Supplementary Fig. 9a, b). The microfluidic device was composed of polydimethylsiloxane (PDMS) layers with microchannels connecting three culture chambers and two medium chambers (Supplementary Fig. 9a). Due to the approximate 1-mm diameter of cardiac tissues fabricated with HEM hydrogel, an inadequate supply of oxygen and nutrients into tissue constructs is inevitable during static culture[36]. Computational simulations of oxygen concentrations in cardiac tissue revealed that dynamic flow with speeds of up to 4.8 cm/sec induced by our microfluidic system facilitates oxygen transfer from the culture medium into the cardiac tissue, while oxygen is relatively scarce in the absence of flow (Fig. 3b, c). Cardiac tissues were assumed to be dense cell spheroids in the simulation, and indeed we found that their overall cell density was consistent and uniform across the whole construct (Supplementary Fig. 9c). Although this simulation analysis was performed for only a short period (~60 mins), it suggests that dynamic flow improves the oxygen supply to the inner cells of cardiac tissues, enabling long-term tissue culture and maturation.

To experimentally validate the effect of flow, we compared the HEM hydrogel-impregnated cardiac tissues cultured in the microfluidic chip with medium flow (HEM-Cf group), a 24-well plate with medium flow (HEM-Pf group), and a 24-well plate under static conditions (HEM-Ps group) for 14 days. First, we assessed oxygen levels in cardiac tissues using a fluorescent hypoxia reagent[17] (Supplementary Fig. 10a). When quantifying the relative fluorescence intensity and area of cardiac tissues in each group, the HEM-Cf group showed significantly lower fluorescence signals than other groups. This demonstrates that dynamic flow in our microfluidic chip can provide sufficient oxygen supply for increased cell viability. Indeed, expression of the apoptotic marker cleaved caspase-3 was much lower in the HEM-Cf group than in the other groups, and filamentous actin (F-actin) indicating the cell cytoskeleton was disrupted in the HEM-Ps group (Fig. 3d). Notably, while the expression of apoptotic cells in HEM-Ps and HEM-Pf group was significantly elevated inside compared with outside of cardiac tissues, the HEM-Cf group exhibited a relatively lower expression of apoptotic cells (Supplementary Fig. 10b, c). Thus, cardiac tissue viability was improved due to dynamic flow in the microfluidic chip. Moreover, the well-type plate groups showed impaired expression of markers for CMs (cardiac troponin T; cTnT), ECs (CD31), and CFs (vimentin; VIM) (Fig. 3e). Clear differences in the structural maturation between the HEM-Ps and HEM-Cf groups were also confirmed through whole-mount staining for each marker

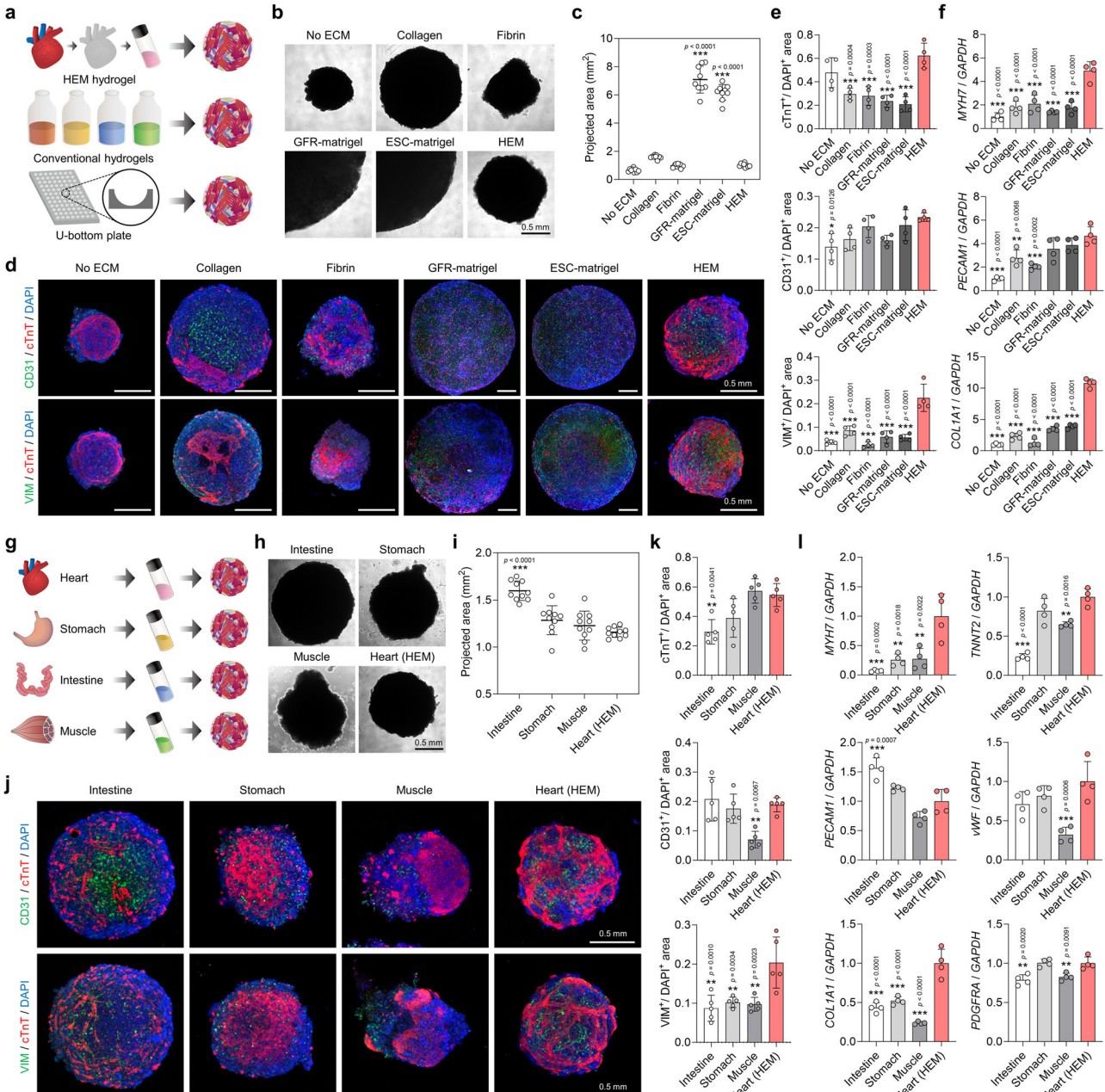

**Fig. 2 | Heart extracellular matrix (HEM) hydrogel is superior to conventional methods for fabricating human cardiac tissues. a** Schematic illustrations of the fabrication of cardiac tissues using HEM hydrogel, conventional commercial hydrogels, or U-bottom plates. **b** Bright-field images and **c** projected areas of human cardiac tissues fabricated using U-bottom plates (No ECM [extracellular matrix]), collagen, fibrin, growth factor reduced (GFR)-Matrigel, embryonic stem cell (ESC)-qualified Matrigel, and HEM hydrogel (scale bar = 0.5 mm, *N* = 9, biological replicates, ***p* < 0.001 versus HEM group). **d** Representative immunofluorescent images and **e** quantification of the areas positive for cardiomyocyte (CM; cTnT), endothelial cell (EC; CD31), and cardiac fibroblast (CF; VIM) markers in each group of cardiac tissues (scale bars = 0.5 mm, *N* = 4, biological replicates, **p* < 0.05 and ****p* < 0.001 versus HEM). The cTnT-positive areas were quantified using immunofluorescent images in (**d**). **f** Quantitative PCR (qPCR) analysis of relative mRNA expression levels of CM (*MYH7*), EC (*PECAM1*), and CF (*COL1A1*) markers in each group of cardiac tissues (*N* = 4, biological replicates, ***p* < 0.01 and ****p* < 0.001 versus HEM group). **g** Schematic illustrations of fabrication of cardiac

tissues using HEM hydrogel and other tissue-derived ECM hydrogels. **h** Bright-field images and **i** projected areas of cardiac tissues fabricated using decellularized intestine, stomach, and muscle-derived ECM hydrogels and HEM hydrogel (scale bar = 0.5 mm, *N* = 10, biological replicates, ****p* < 0.001 versus HEM group). **j** Representative immunofluorescent images and **k** quantification of the areas positive for CM (cTnT), EC (CD31), and CF (VIM) markers in each group of cardiac tissues (scale bars = 0.5 mm, *N* = 5, biological replicates, ***p* < 0.01 versus HEM group). The cTnT-positive areas were quantified using immunofluorescent images in (**j**). **l** qPCR analysis of relative mRNA expression levels of CM (*MYH7* and *TNNT2*), EC (*PECAM1* and *vWF*), and CF (*COL1A1* and *PDGFRA*) markers in each group of cardiac tissues (*N* = 4, biological replicates, ***p* < 0.01 and ****p* < 0.001 versus HEM group). Cardiac tissues prepared with human induced pluripotent stem cell (hiPSC)-derived CMs, CFs, and human umbilical vein endothelial cells (HUVECs) were cultured for 7 days and used for analysis. Data are presented as means ± S.D. Statistical significance was determined using one-way ANOVA with Tukey's multiple comparisons tests in (**c, e, f, i, k,** and **l**).

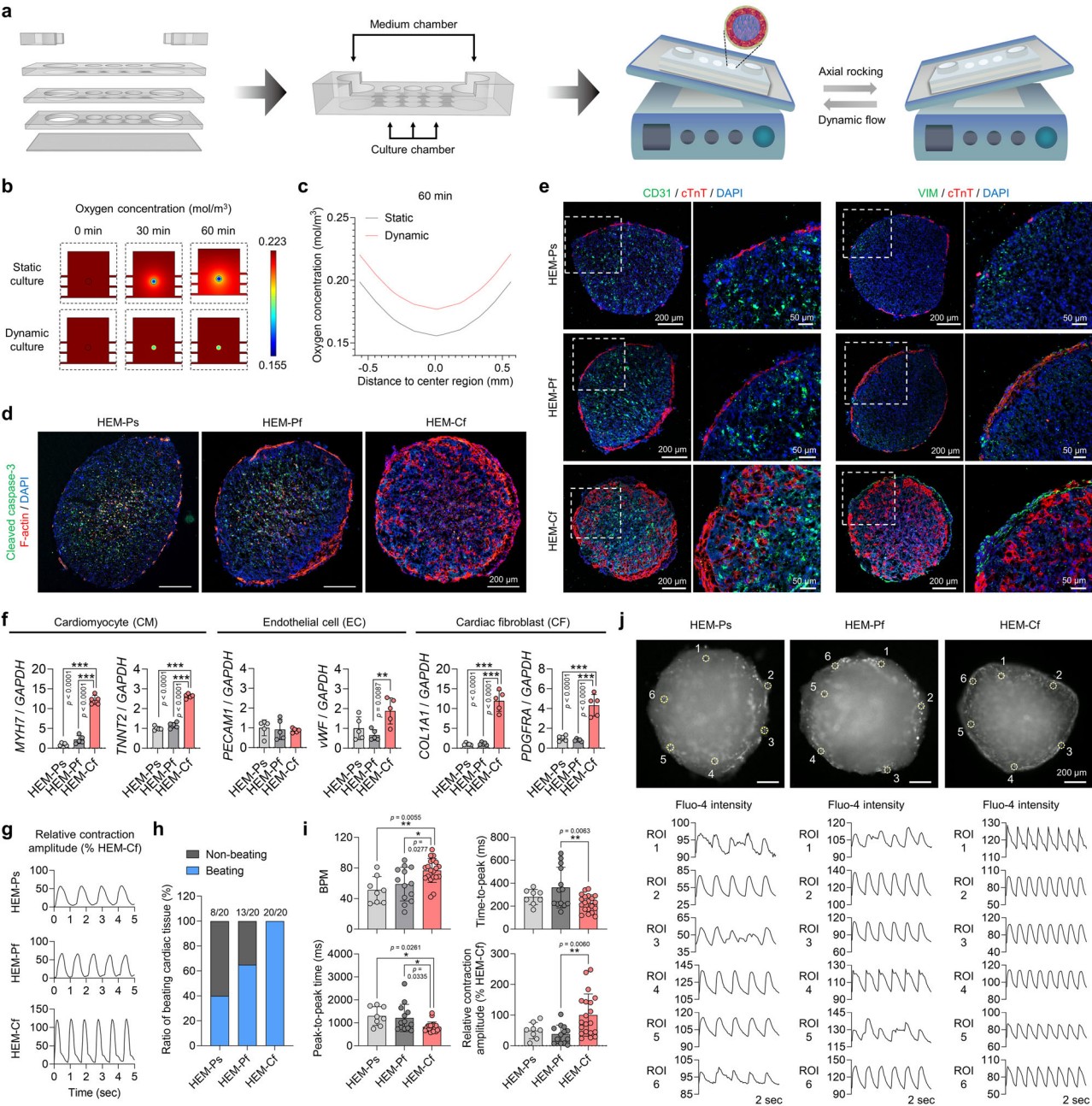

**Fig. 3 | The effects of dynamic flow in microfluidic chips on human cardiac tissues. a** Schematic illustrations of cardiac tissues cultured with medium dynamic flow in microfluidic chips on an axial rocker. **b** Simulation analysis of oxygen concentrations in cardiac tissues cultured in microfluidic chips under static or dynamic conditions. **c** Simulated oxygen profiles in the cardiac tissues for 60 min under each condition. **d** Immunofluorescent images of cleaved caspase-3 and F-actin in cardiac tissues fabricated using heart extracellular matrix (HEM) hydrogel and then cultured in 24-well plates without flow (HEM-Ps; Plate static), 24-well plates with flow (HEM-Pf; Plate flow), and chips with flow (HEM-Cf; Chip flow) (scale bars = 200 μm, representative images from two independent experiments). **e** Immunofluorescent images of cardiomyocyte (CM; cTnT), endothelial cell (EC; CD31), and cardiac fibroblast (CF; VIM) markers in each group of cardiac tissues (scale bars = 200 μm for the left images and 50 μm for the right images). **f** Quantitative PCR (qPCR) analysis of relative mRNA expression levels of CM (*MYH7* and *TNNT2*), EC (*PECAM1* and *vWF*), and CF (*COL1A1* and *PDGFRA*) markers in each group of cardiac tissues (*N* = 5, biological replicates, **p < 0.01 and ***p < 0.001, representative data from

two independent experiments). **g** Representative contraction traces of spontaneously beating cardiac tissues. Contraction amplitude in each group was normalized to the mean value of the HEM-Cf group. **h** Ratio of spontaneously beating cardiac tissues in each group. **i** Beats per minute (BPM), peak-to-peak time, time-to-peak, and relative contraction amplitude values were calculated from the values analyzed using MUSCLEMOTION (*N* = 8 for HEM-Ps, *N* = 13 for HEM-Pf, and *N* = 20 for HEM-Cf, biological replicates, *p < 0.05 and **p < 0.01). Quantitative data were analyzed from two independent experiments. **j** Fluorescent images of Fluo-4 (calcium indicator) in each group of spontaneously beating cardiac tissues (scale bars = 200 μm) and graphs of calcium traces obtained from the six regions of interest (ROIs) (representative data from two independent experiments). Cardiac tissues prepared with human induced pluripotent stem cell (hiPSC)-derived CMs, CFs, and human umbilical vein endothelial cells (HUVECs) were cultured for 14 days. Data are presented as means ± S.D. Statistical significance was determined using one-way ANOVA with Tukey's multiple comparisons tests in (**f**) and Bonferroni's multiple comparisons tests in (**i**).

(Supplementary Fig. 11). During the monitoring period from day 4 to day 21, the expression of cTnT gradually decreased in the HEM-Ps group, whereas it was maintained in the HEM-Cf group (Supplementary Fig. 12). These results support our hypothesis that dynamic flow is essential for long-term culture of macroscale cardiac tissues.

We investigated the effects of dynamic flow on the cardiovascular development and functionality of human cardiac tissues (Fig. 3f–j). The expression levels of CM, EC, and CF markers were higher in the HEM-Cf group than in the control groups (Fig. 3f). The cardiac tissues cultured with dynamic flow in the microfluidic chip exhibited functional improvements, as evidenced by the strongest and fastest contractility in the HEM-Cf group (Fig. 3g). All cardiac tissues in the HEM-Cf group displayed autonomous beating, but 40–65% of the cardiac tissues did not show spontaneous beating in the other groups (Fig. 3h). The absence of dynamic flow in the microfluidic chip resulted in a significant reduction in beats per minute (BPM) and an increase in the peak-to-peak time of contraction (Fig. 3i). The time-to-peak also increased in both well-type plate groups, which could be interpreted as an indication of immaturity or a phenotype of heart disease[37]. Interestingly, the BPM of cardiac tissues in the HEM-Cf group was 76.59 ± 15.7 (Fig. 3i), which falls within the normal human heartbeat range[38]. In addition, the contraction amplitude, an indicator of CM maturation, was significantly higher in the HEM-Cf group than in the other groups[39]. Synchronized contraction of the cardiac tissues was assessed by calcium imaging using Fluo-4 (Fig. 3j and Supplementary Movies 1, 2). In the HEM-Cf group, synchronized beats were observed in all six regions of interest (ROI), while both well-type plate groups (HEM-Ps and HEM-Pf groups) exhibited irregular beats in some regions of interest. Our findings suggest that microfluidic chip-based flow can improve the functional maturation of human cardiac tissue.

## Genetic and structural signatures associated with the maturation of cardiac tissues

To isolate the effects of HEM hydrogel and dynamic flow on the maturation of cardiac tissues, four groups with different combinations of the two factors (No HEM-Ps, HEM-Ps, No HEM-Cf, and HEM-Cf) were analyzed using RNA sequencing to compare their overall transcriptomic profiles. PCA showed that each group had a distinct profile and the samples in each group could be clustered (Fig. 4a). When the differentially expressed genes (DEGs) between the groups were manifested as a heatmap, the effects of dynamic flow on the transcriptional profiles were more prominent (Fig. 4b). For a more detailed comparison, the DEGs between the HEM-Cf group and the other three groups were displayed as volcano plots (Supplementary Fig. 13a–c). The HEM-Cf group exhibited a greater number of DEGs compared to the No HEM-Ps group (Supplementary Fig. 13a), indicating that the presence of both HEM hydrogel and dynamic flow may have significant impact on the transcriptional profiles of cardiac tissues. We conducted Gene Ontology (GO) analysis focusing on the upregulated DEGs and examined the associated GO terms (Fig. 4c, d). Several genes associated with heart development, ECM organization, and angiogenesis were significantly upregulated in the HEM-Cf group, which is likely to contribute to the enhanced differentiation and functional improvement of the cardiac tissues (Fig. 3).

We also conducted GO analysis focusing on the effects of HEM in improving drug testing and disease modeling. Several GO terms, including response to drug, response to cytokine, and arrhythmogenic right ventricular cardiomyopathy were significantly upregulated in the HEM-Cf group compared to the No HEM-Cf group (Supplementary Fig. 14a). The GO terms related to collagen and ECM conferring elasticity and tensile strength was also upregulated in the HEM-Cf group. This indicates that HEM hydrogel can provide a suitable microenvironment for drug testing and disease modeling associated with arrhythmias and fibrosis. Specifically, *SPARC*, *SFRP2*, *GREM1*, *FOSL1*, *FBN1*, *FBLN5*, *COL1A1*, and *COL3A1* were significantly upregulated by

the incorporation of HEM hydrogel (Supplementary Fig. 14b, c). SPARC[40,41] and SFRP2[42,43] are known to mediate collagen deposition and fibrosis, and increased FBN1 deposition is accompanied by fibrosis progression[44]. Moreover, SFRP2 treatment is known to promote the maturation of iPSC-derived CMs[45], and *FBN1* gene is considered as one of the markers for mature CM phenotypes[46]. The elevated expression of these genes in cardiac tissues fabricated with HEM hydrogel could support the deliberate cardiac fibrosis modeling and precise response to drug through improved maturity. The upregulated expression of four genes (*SPARC*, *SFRP2*, *FOSL1*, and *FBN1*) was verified by qPCR analysis (Supplementary Fig. 14d).

When we compared the sarcomere structures in cardiac tissues cultured for 14 days, the HEM-Cf group exhibited the most aligned and longest sarcomere structures (Fig. 4e). Specifically, the average sarcomere length in the HEM-Cf group measured 1.87 ± 0.14 μm, a level similar to that seen in late-stage CMs and other cardiac models like engineered heart tissues (EHT)[47,48]. Moreover, high-magnification images of cTnT staining showed that individual CMs having different nuclei were structurally interconnected with well-developed sarcomeres (Fig. 4f). Alongside the synchronized $Ca^{2+}$ transients (Supplementary Movie 2), these data provide evidence that cardiac tissues of HEM-Cf group can function as a syncytium.

In our study, the combination of 3D HEM hydrogel and microfluidic flow promoted genetic signatures of human cardiac tissues (human microfluidic HEM cardiac tissue; μF/HEM-CT), as well as enhanced phenotypic and structural maturation. The gene expression levels of the CM marker *TNNT2* and the cardiac ion channels *SCN5A* and *CACNA1C* were observed to gradually increase up to 21 days during the 3D HEM culture in the microfluidic chip (Fig. 4g). Transitions in cardiac gene expression of *TNNI1* to *TNNI3*, *MYL7* to *MYL2*, and *MYH6* to *MYH7* are recognized as key changes in the maturation process of CMs[49–52]. The ratios of *TNNI3/TNNI1*, *MYL2/MYL7*, and *MYH7/MYH6* were significantly upregulated in the μF/HEM-CT over the course of 21 days (Fig. 4g), indicating that the human cardiac tissues underwent cardiac maturation during the 3D HEM hydrogel culture in the microfluidic chip.

Cardiac tissue organization and structural maturity were improved in the μF/HEM-CT by day 21 of culture (Fig. 4h–l). Interestingly, CMs, ECs, and CFs were compartmentalized in the tissue constructs (Fig. 4h). Expression of gap junctions was verified by connexin-43 (CX43) staining (Fig. 4i). Serial section scanning electron microscopy was applied to observe the ultrastructure of the μF/HEM-CT (Fig. 4j–l). The presence of intercellular junctions between CMs and CFs was confirmed in low-magnification images (Fig. 4j). Organelles abundantly observed in muscle cells, such as myofibril and mitochondria, were identified inside CMs, and intercalated discs were observed at junctions between the CMs (Fig. 4k). Microstructures, including Z-line, I-band, and A-band, in the myofibril were also observed in high-magnification images (Fig. 4l). These data validate that our culture platform can create human cardiac tissues with matured cardiac-specific microstructures.

## Establishment of a human cardiac tissue-based drug evaluation platform

We confirmed the feasibility of μF/HEM-CT with high maturity as a platform for drug evaluation. Epinephrine and isoproterenol, which increase beat frequency[7,53], were used to assess drug responses in cardiac tissues. Based on our contraction analysis, we observed a decrease in peak-to-peak time and an increase in BPM after drug treatment (Fig. 5a, b). We assessed the alteration in contractile phenotypes through treatment with nifedipine, which is an L-type calcium channel blocker[54]. When μF/HEM-CT was treated with various nifedipine concentrations (0.01-1 μM), the contraction amplitude diminished in a dose-dependent manner (Fig. 5c). Similarly, peak shortening, contraction velocity, and relaxation velocity decreased with an

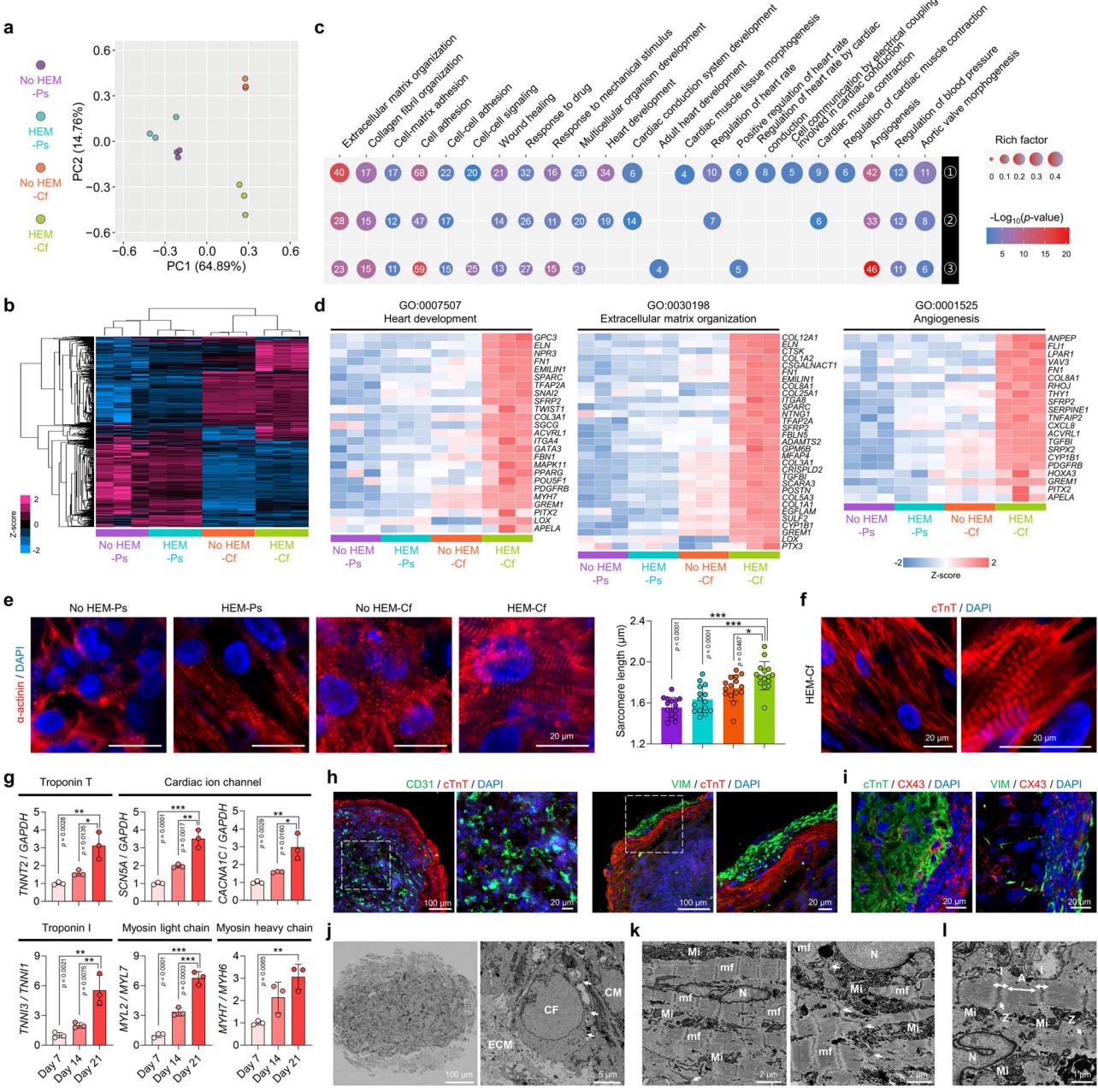

increase in drug doses (Fig. 5c). Changes in electrophysiological properties of the µF/HEM-CT following drug treatments were also measured using a multi-electrode array (MEA). When the cardiac tissues were treated with E-4031, which inhibits the potassium channel of the hERG (human ether-à-go-go-related gene)-type, the prolongation of field potential duration corrected by Fridericia's formula ($FPD_{cf}$) was observed (Fig. 5d). This effect was intensified as the drug concentration increased, while no changes were observed in BPM or field potential amplitude (FPA). Changes in electrical properties in response to nifedipine were also detected in the cardiac tissues, including shortened $FPD_{cf}$ and increased BPM (Fig. 5e). These data indicate that the human cardiac tissues in the 3D microfluidic HEM culture (µF/HEM-CT) provide adequate models for evaluating drug-induced contractile patterns and electrophysiological changes.

We assessed whether our µF/HEM-CT could detect unexpected cardiotoxicity, which is a major hurdle to the development of new drugs[55]. Drug-induced blocking of ion channels is one of the main mechanisms of cardiotoxicity. Specifically, the inhibition of hERG

channels can result in increased action potential duration, prolongation of QT intervals, and an increased risk of Torsades de Pointes (TdP)[55,56]. We investigated the responses of cardiac tissues to 8 drugs among 28 drugs that evoke clinical TdP risk[57]. Three drugs in the high TdP risk category (vandetanib, bepridil, and quinidine), three drugs in the intermediate TdP risk category (chlorpromazine, terfenadine, and clarithromycin), and two drugs in the low TdP risk category (diltiazem and ranolazine) were evaluated in a dose-dependent manner (Fig. 5f). Drug concentration-dependent changes in the electrical properties of human cardiac tissues were consistent with previously reported results using hiPSC-CMs[57–59]. Among the five drugs in the intermediate and low TdP risk categories, four drugs (chlorpromazine, terfenadine, clarithromycin, and ranolazine) did not induce significant changes in $FPD_{cf}$, while increasing concentrations of diltiazem resulted in decreased $FPD_{cf}$ (Fig. 5f). Treatment with vandetanib and quinidine, both in the high TdP risk category, resulted in the extension of $FPD_{cf}$ in cardiac tissues as their concentrations increased, eventually leading to the

**Fig. 4 | Transcriptomic profiles and structural maturity of human cardiac tissues. a** Principal component analysis plot and **b** hierarchical clustering heatmap of differentially expressed genes (DEGs) between cardiac tissues fabricated using U-bottom plates (No HEM [heart extracellular matrix]) or HEM hydrogel (HEM) and then cultured in 24-well plates without flow (Ps) or chips with flow (Cf). Four groups were generated according to the fabrication and culture methods, and transcriptomic profiles were compared after 14 days of culture ($N = 3$, biological replicates). **c** Dot plot showing Gene Ontology (GO) terms for upregulated DEGs in the HEM-Cf group, compared to ① No HEM-Ps, ② HEM-Ps, and ③ No HEM-Cf groups. Gene counts are presented in the circles, and rich factors were calculated by dividing the GO-annotated DEGs by all GO-annotated genes. Statistical test was conducted using Cuffdiff program and DEGs were selected with the following parameters: fold change > 2, $p$ value < 0.05, and FDR < 0.1. **d** Heatmaps displaying upregulated DEGs from three GO terms: heart development (GO:0007507), extracellular matrix organization (GO:0030198), and angiogenesis (GO:0001525) ($N = 3$, biological replicates). **e** Representative immunofluorescent images of sarcomere structure (α-actinin) and quantification of sarcomere length in each group of cardiac tissues (scale bars = 20 μm, $N = 15$, biological replicates, $*p < 0.05$ and $***p < 0.001$). **f** Representative immunofluorescent images of sarcomere structure (cTnT) in cardiac tissues (scale bars = 20 μm, $N = 4$, biological replicates). **g** Quantitative PCR (qPCR) analysis of relative mRNA expression levels related to troponin T (TNNT2), cardiac ion channels (SCN5A and CACNA1C), troponin I (TNNI3/

TNNI1), myosin light chain (MYL2/MYL7), and myosin heavy chain (MYH7/MYH6) in cardiac tissues cultured for 7, 14, or 21 days ($N = 3$, biological replicates, $*p < 0.05$, $**p < 0.01$, and $***p < 0.001$). **h** Representative immunofluorescent images of cardiomyocytes (CM; cTnT), endothelial cell (EC; CD31), and cardiac fibroblast (CF; VIM) markers in cardiac tissues (scale bars = 100 μm for left images and 20 μm for right images, $N = 1$ from three independent experiments). **i** Representative immunofluorescent images of CM (cTnT), CF (VIM), and gap junction (CX43) markers in cardiac tissues (scale bars = 20 μm, $N = 1$ from two independent experiments). **j** Field emission-scanning electron microscopic (FE-SEM) images showing sections of cardiac tissues. White arrows indicate intercellular junctions between CM and CF (scale bar = 100 μm for the left image and 5 μm for the right image, $N = 1$ from two independent experiments). **k** FE-SEM images showing myofibrils (mf), mitochondria (Mi), and nuclei (N) of CMs in cardiac tissues. White arrows indicate intercalated discs between CMs (scale bars = 2 μm). **l** FE-SEM image showing Z-lines (Z), I-bands (I), A-bands (A), mitochondria (Mi), and nuclei (N) of CMs in cardiac tissues (scale bar = 1 μm). Cardiac tissues prepared using human induced pluripotent stem cell (hiPSC)-derived CMs, CFs, and human umbilical vein endothelial cells (HUVECs) were used for the analyzes. The cardiac tissues in (**f**–**l**) are samples from the HEM-Cf group, and cardiac tissues in (**e**–**f**) and (**h**–**l**) were cultured for 14 and 21 days, respectively. Data are presented as means ± S.D. and statistical significance was determined using one-way ANOVA with Tukey's multiple comparisons tests in (**e** and **g**).

cessation of field potential. Conversely, no prolongation of $FPD_{cf}$ was observed in cardiac tissues treated with bepridil, which is consistent with a previous study[60]. Most groups showed no change in BPM, with only diltiazem leading to an increase in BPM (Fig. 5f). Diltiazem was also the only drug that induced an FPA increase, while bepridil, quinidine, chlorpromazine, and terfenadine all reduced the FPA; no significant changes in FPA were observed in tissues treated with vandetanib, clarithromycin, or ranolazine (Fig. 5f). Thus, human cardiac tissues integrated with the multi-electrode array could enable simultaneous detection of alterations in various parameters caused by drugs. These results collectively demonstrate the feasibility of our cardiac tissues for drug cardiotoxicity evaluations.

## Establishment of heart disease models that recapitulate human pathophysiology

As another application of μF/HEM-CT fabricated by 3D microfluidic HEM hydrogel culture, we established two in vitro disease models for Long QT Syndrome (LQTS) and cardiac fibrosis by exploiting the physiological similarity of μF/HEM-CT to native human heart tissue. LQTS is an inherited arrhythmic disease caused by mutations in cardiac ion channels. The types of LQTS from LQT1 to LQT15 are determined by which ion channel is problematic[61]. LQT2, caused by loss-of-function mutations in the KCNH2 gene encoding the hERG channel, is one of the three major types (LQT1−3) that account for 90% of genetically confirmed LQTS patients[62].

The development of in vitro LQTS tissue models has primarily focused on CMs alone[63–65], and the effect of CFs on LQTS disease remains unclear. Therefore, we assessed the respective roles of CMs and CFs in LQTS modeling by utilizing our cardiac tissues. For comparison studies, we prepared human cardiac tissues by combining either normal iPSC-derived CMs and CFs or LQT2 patient iPSC-derived LQT2-CMs and LQT2-CFs (Fig. 6a and Supplementary Fig. 15). Cardiac tissues from each combination exhibited similar expression levels of CM and CF markers (cTnT and VIM, respectively) and compartmentalization of CMs and CFs (Fig. 6b). However, when contraction analysis was conducted under 1-Hz electrical pacing conditions, cardiac tissues composed of LQT2-CMs and LQT2-CFs showed relatively prolonged relaxation unlike the other groups that showed normal waveforms (Fig. 6c). When several parameters of contraction were quantified, the relaxation time was longer in the cardiac tissues composed of LQT2 patient-derived cells. Moreover, the ratio of relaxation time to contraction duration was also significantly higher in cardiac

tissues containing LQT2-CMs and LQT2-CFs than in normal tissues (Fig. 6d). Then, we assessed the electrophysiological characteristics of LQTS cardiac tissue models using MEA (Fig. 6e, f). The cardiac tissues fabricated with LQT2-CMs and LQT2-CFs exhibited significantly prolonged FPD when compared with normal cardiac tissues. Moreover, cardiac tissues fabricated with either LQT2-CMs and normal CFs or normal CMs and LQT2-CFs also showed prolonged FPD compared to the normal cardiac tissues. However, their FPD prolongations were shorter than those of cardiac tissues fabricated using both LQT2-CMs and LQT2-CFs. These results suggest that while either LQT2-CMs or LQT2-CFs can serve individually to depict LQTS traits, their collaborative utilization results in a more realistic representation of LQTS characteristics.

We also evaluated the suitability of μF/HEM-CT for in vitro cardiac fibrosis modeling. Fibrotic changes in cardiac tissues were induced by extrinsic transforming growth factor-β1 (TGF-β1) (Fig. 6g). Cardiac fibrosis, characterized by excessive ECM production, can progress to heart failure due to increased stiffness and deterioration of contractile function[66]. Activation of the fibrotic pathway is primarily mediated by TGF-β signaling[66,67]. Upon treatment with 50 ng/ml TGF-β1, no significant morphological changes were observed in the cardiac tissues (Fig. 6h). However, disruption of the CM structure and an increase in the CF portion were detected (Fig. 6i), which were likely due to fibrotic activation. Masson's trichrome (MT) staining showed collagen deposition and significant increase of the collagen⁺ fibrotic area in our fibrosis model (Fig. 6j). In the case of contraction profiles, the fibrosis model showed a significant decrease in contraction amplitude, contraction duration, and peak-to-peak time, while the BPM increased (Fig. 6k). The decreased contraction amplitude and increased BPM in the fibrotic cardiac tissues are consistent with the pathophysiological features of previously reported fibrotic tissue models[18,68,69]. Then, we conducted drug testing using losartan, an angiotensin II type 1 receptor antagonist known for its clinical efficacy in regressing myocardial fibrosis[70,71]. The anti-fibrotic effects of this drug were previously demonstrated in human cardiac fibrosis-on-a-chip model[72]. In our study, the fibrotic tissues were treated with 5 μM losartan on day 14 and cultured for additional 7 days (Fig. 6l). The disrupted CM structure in the fibrosis model was restored by the treatment of losartan (Fig. 6m, n). The impaired contraction profiles of the fibrosis model were also recovered by losartan treatment (Fig. 6o). These results suggest that cardiac tissues established in our culture platform (μF/HEM-CT) can serve as a valuable cardiac fibrosis model for evaluation of anti-fibrotic efficacy of the drugs.

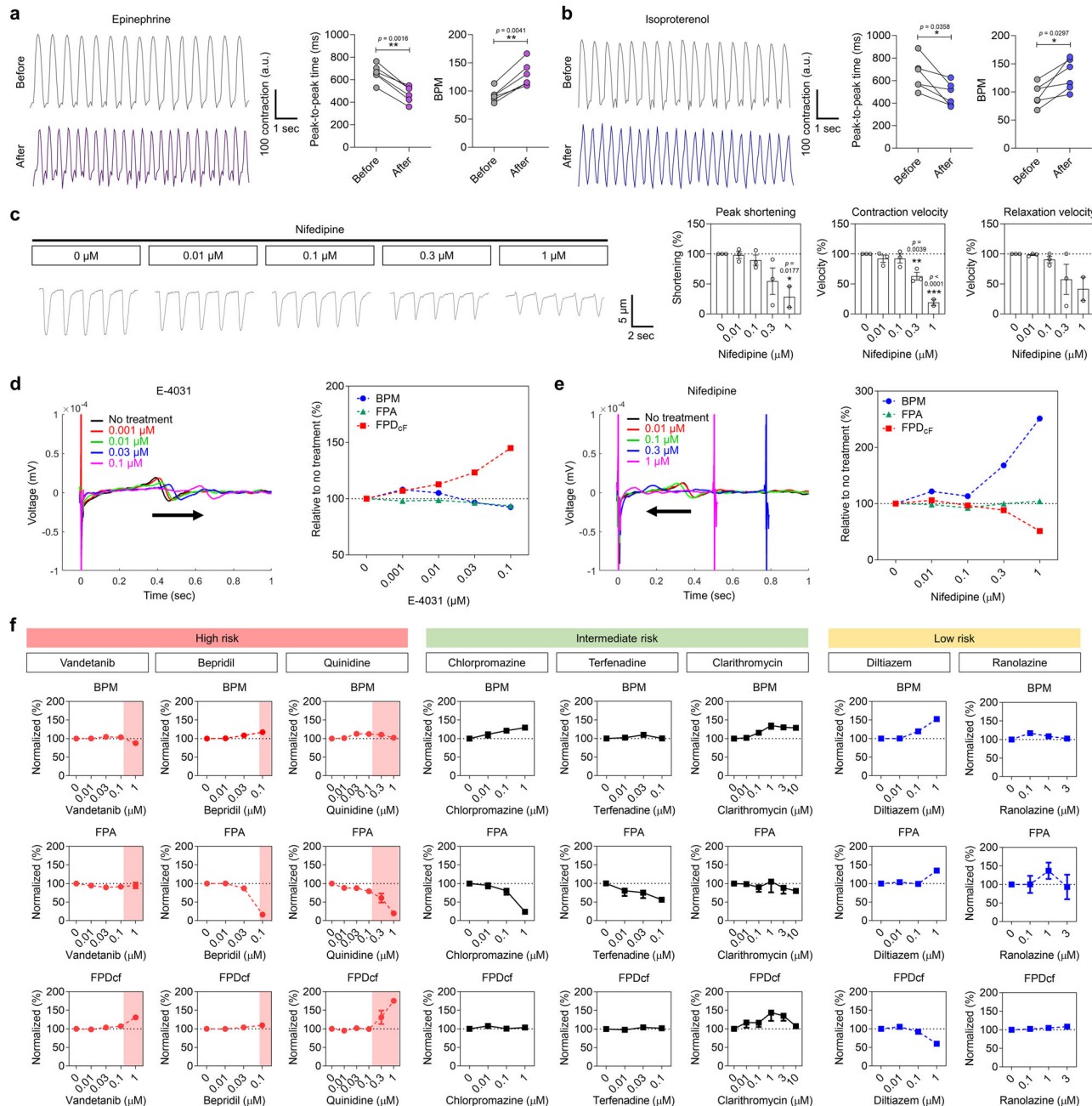

**Fig. 5 | Application of human cardiac tissues to evaluate drug responses and cardiotoxicity.** Representative contraction traces and contraction parameters related to the beat frequency of cardiac tissues before and after treatment with **a** epinephrine and **b** isoproterenol. The peak-to-peak time and beats per minute (BPM) were calculated from the values analyzed using MUSCLEMOTION ($N = 6$, biological replicates, $*p < 0.05$ and $**p < 0.01$). **c** Representative contraction traces and contraction parameters (peak shortening, contraction velocity, and relaxation velocity) of cardiac tissues following nifedipine treatment at concentrations of 0, 0.01, 0.1, 0.3, and 1 μM ($N = 2$ for 1 μM and $N = 3$ for the other concentrations, biological replicates, $*p < 0.05$, $**p < 0.01$, and $***p < 0.001$ versus 0 μM) as analyzed using SoftEdge™ Acquisition software. Representative action potential traces and

action potential parameters (BPM, field potential amplitude [FPA], and Fridericia's formula [FPD$_{cF}$]) of cardiac tissues following **d** E-4031 treatment (0, 0.001, 0.01, 0.03, and 0.1 μM; $N = 3$, biological replicates) and **e** nifedipine treatment (0, 0.01, 0.1, 0.3, and 1 μM; $N = 3$, biological replicates). **f** The respective action potential parameters (BPM, FPA, and FPD$_{cf}$) of cardiac tissues after treatment with various concentrations of eight drugs belonging to three risk categories of Torsades de Pointes (TdP; $N = 3$, biological replicates). All cardiac tissue samples were prepared in microfluidic heart extracellular matrix (HEM) hydrogel conditions with dynamic flow (HEM-Cf group). Data are presented as means ± S.E.M. Statistical significance was determined using unpaired two-sided Student's $t$-tests in (**a** and **b**) and one-way ANOVA with Dunnett's multiple comparisons tests in (**c**).

## Transplantation of human cardiac tissues as regenerative medicine for myocardial infarction

We assessed the therapeutic potential of our cardiac tissues for cardiac regeneration in a rat model of myocardial infarction. We transplanted μF/HEM-CT into infarcted hearts one week after inducing ischemia/reperfusion injury by ligating the left anterior descending (LAD) artery for 1 h (Fig. 7a). Cardiac tissues

for transplantation were prepared using red fluorescence protein (RFP)-expressing hiPSCs to efficiently track the grafted cells in vivo. The RFP-hiPSCs, of which more than 98% expressed OCT4, were differentiated into RFP-CMs (Supplementary Fig. 16a, b). Cardiac tissues were created by co-culturing RFP-CMs with ECs and hiPSC-CFs, resulting in the presence of cTnT⁺ RFP⁺ cells (Supplementary Fig. 16c, d).

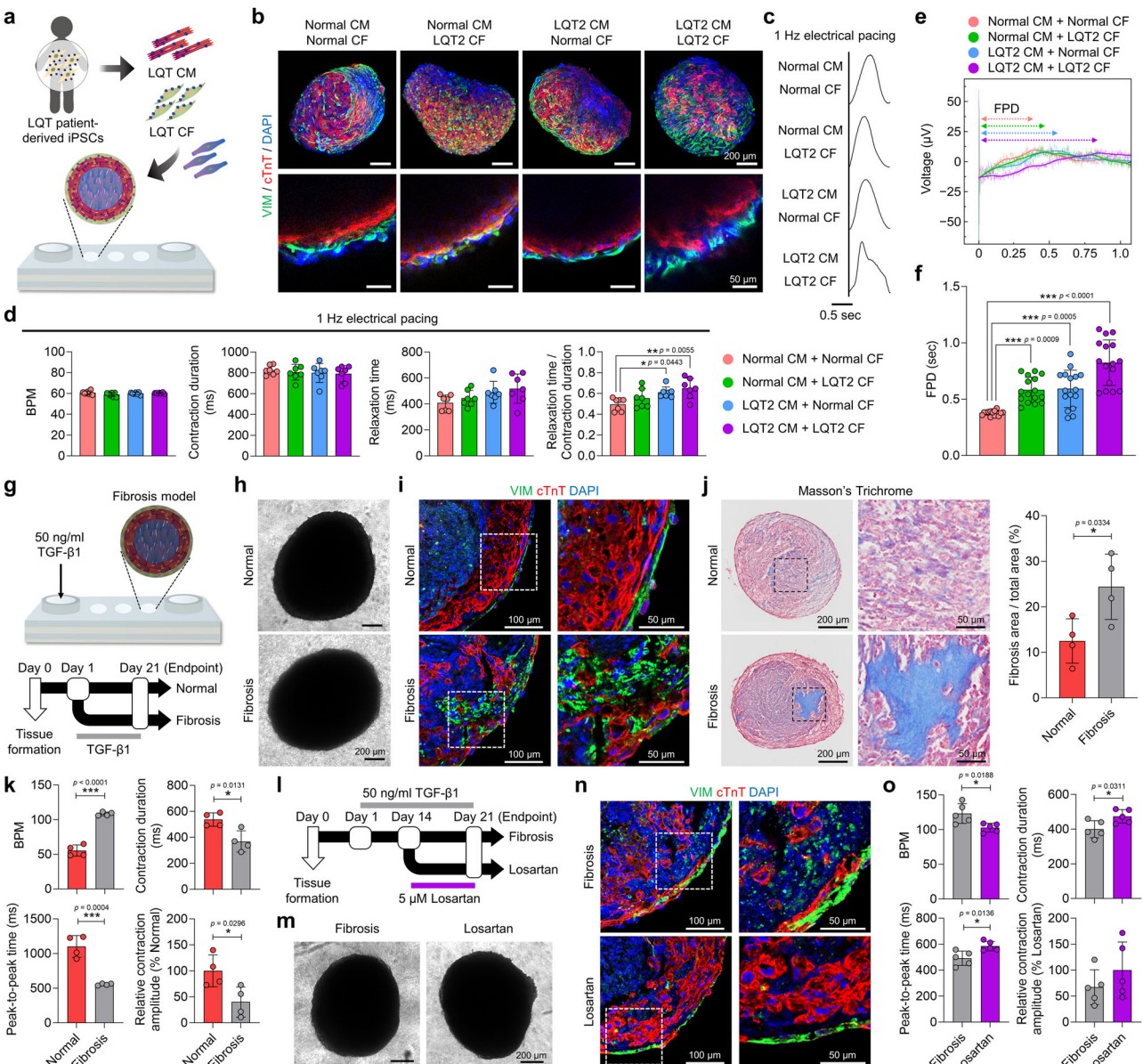

**Fig. 6 | Modeling of heart diseases using human cardiac tissues. a** Schematic illustration of the generation of Long QT Syndrome (LQTS) cardiac tissues using cardiomyocytes (CMs) and cardiac fibroblasts (CFs) differentiated from LQTS patient-derived induced pluripotent stem cells (iPSCs) in microfluidic chips. **b** Representative immunofluorescent images of CM (cTnT) and CF (VIM) markers in cardiac tissues fabricated using normal CMs or LQT2 CMs and normal CFs or LQT2 CFs (scale bars = 200 μm for upper images and 50 μm for lower images, representative images from two independent experiments). **c** Representative contraction traces of cardiac tissues in each group with 1-Hz electrical pacing. **d** Contraction parameters (beats per minute [BPM], contraction duration, relaxation time, and relaxation time/contraction duration) of cardiac tissues in each group with 1-Hz electrical pacing ($N = 7$, biological replicates, *$p < 0.05$ and **$p < 0.01$, quantitative data analyzed from two independent experiments). **e** Representative field potential traces and **f** quantification of field potential duration (FPD) of cardiac tissues in each group ($N = 16$ from 4 biological replicates, ***$p < 0.001$). **g** Schematic illustration and experimental timeline of the generation of fibrotic cardiac tissues through TGF-β1 treatment. **h** Bright-field images of normal cardiac tissues and fibrosis-induced cardiac tissues (scale bars = 200 μm, $N = 4$, biological replicates). **i** Immunofluorescent images of CM (cTnT) and CF (VIM) markers in sections of normal and fibrotic cardiac tissues (scale bars = 100 μm for the left images and 50 μm for the right images, $N = 4$, biological replicates).

**j** Masson's trichrome (MT) staining of normal and fibrotic cardiac tissues and quantification of blue⁺ fibrosis area (%) relative to total area (scale bars = 200 μm for left images and 50 μm for right images, $N = 4$, biological replicates, *$p < 0.05$, representative data from two independent experiments). **k** The BPM, peak-to-peak time, contraction duration, and relative contraction amplitude in each group of spontaneously beating cardiac tissues, were calculated using MUSCLEMOTION ($N = 4$, biological replicates, *$p < 0.05$ and ***$p < 0.001$). **l** Experimental timeline of drug testing using fibrotic cardiac tissues and anti-fibrotic drug (losartan). **m** Bright-field images of fibrosis-induced and drug-treated cardiac tissues (scale bars = 200 μm, $N = 5$, biological replicates). **n** Immunofluorescent images of CM (cTnT) and CF (VIM) markers in sections of fibrotic and drug-treated cardiac tissues (scale bars = 100 μm for the left images and 50 μm for the right images, $N = 5$, biological replicates). **o** The BPM, peak-to-peak time, contraction duration, and relative contraction amplitude in each group of spontaneously beating cardiac tissues, calculated using MUSCLEMOTION ($N = 5$, biological replicates, *$p < 0.05$). All cardiac tissues analyzed in Fig. 6 were cultured under microfluidic heart extracellular matrix (HEM) conditions (HEM-Cf group) for 21 days. Data are presented as means ± S.D. Statistical significance was determined using unpaired two-sided Student's *t*-tests in (**j**, **k**, and **o**) and one-way ANOVA with Tukey's multiple comparisons *t*ests in (**d** and **f**).

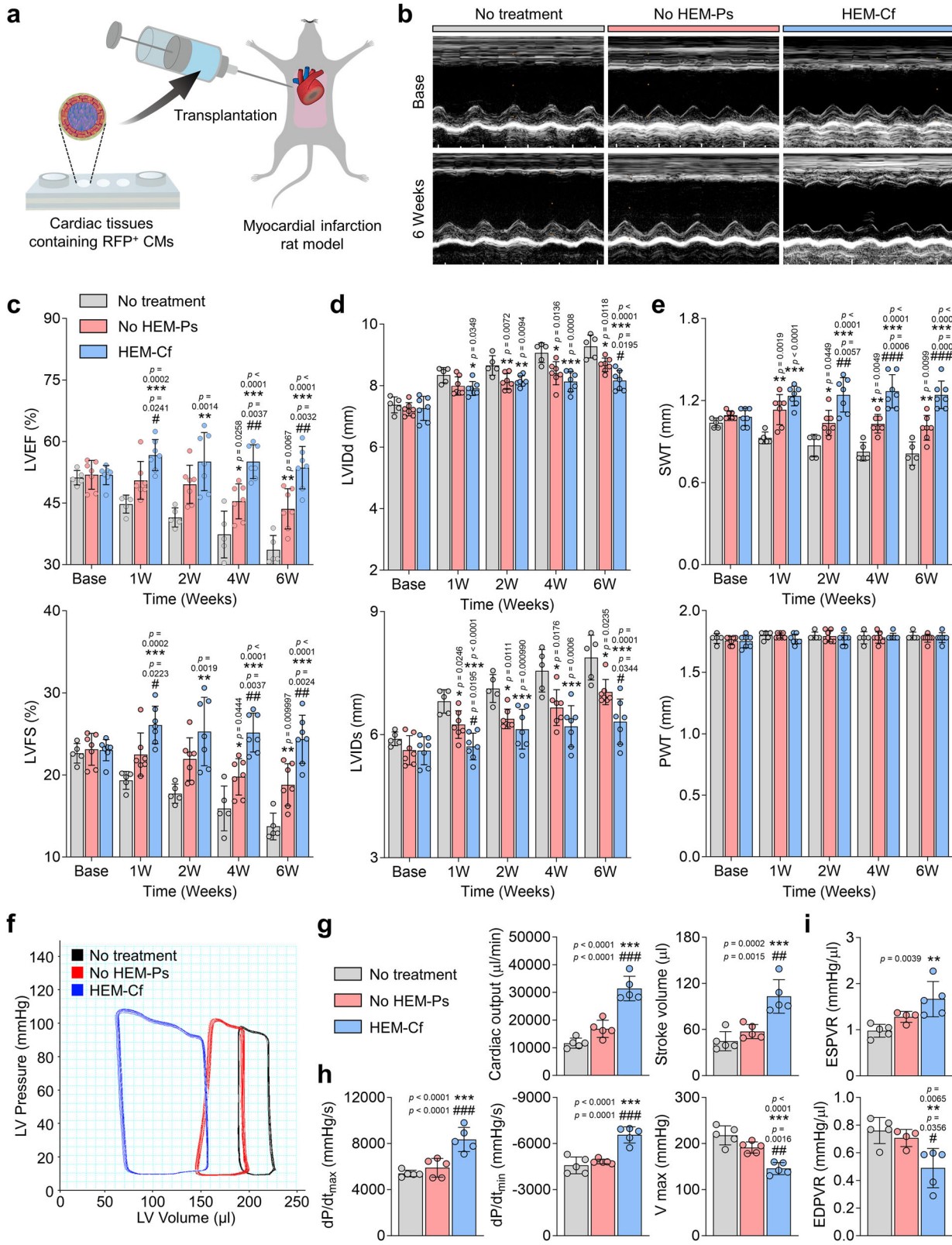

We compared human cardiac tissues generated in the HEM-Cf group (μF/HEM-CT) to those in the No HEM-Ps group. Serial echocardiography was conducted at 1, 2, 4, and 6 weeks after transplantation, and the results revealed that the left ventricular ejection fraction (LVEF) and fractional shortening (LVFS) were significantly higher in the HEM-Cf group than in the other two groups (No treatment and No HEM-Ps) at 1 week, and this trend continued for up to 6 weeks

(Fig. 7b, c). Furthermore, the left ventricular internal diameters at end-diastolic (LVIDd) and end-systolic (LVIDs) phases were significantly smaller in the HEM-Cf group compared to the other groups at 6 weeks, indicating that the cardiac tissues of the HEM-Cf group prevented adverse cardiac remodeling processes (Fig. 7d). The septal wall thickness (SWT) was thicker in the HEM-Cf group compared to the other groups, and the posterior wall thickness (PWT) showed no significant

**Fig. 7 | Therapeutic efficacy of human cardiac tissue transplantation for improving cardiac function after myocardial infarction. a** Schematic illustration of transplantation of cardiac tissues containing human induced pluripotent stem cell (hiPSC)-derived RFP+ cardiomyocytes (CMs) in a rat model of myocardial infarction induced by left anterior descending (LAD) ligation and reperfusion. **b** Representative echocardiography images of M-mode in three groups at baseline (1 week after myocardial infarction) and 6 weeks post-transplantation. **c** Quantification of left ventricular ejection fraction (LVEF), left ventricular fractional shortening (LVFS), **d** left ventricular internal diameter end-diastole (LVIDd), left ventricular internal diameter end-systole (LVIDs), **e** septal wall thickness (SWT), and posterior wall thickness (PWT) values ($N = 5$ for No treatment and $N = 7$ for No HEM-Ps and HEM-Cf, biological replicates, *$p < 0.05$, **$p < 0.01$, and ***$p < 0.001$ versus No treatment group, #$p < 0.05$, ##$p < 0.01$, and ###$p < 0.001$ versus No HEM-Ps group). **f** Representative graph of the hemodynamic pressure and volume (PV) curve during steady state at 4 weeks post-transplantation. **g** Quantitative graphs of cardiac output, stroke volume, **h** maximal rate of pressure changes during systole ($dP/dt_{max}$), minimal rate of pressure changes during diastole ($dP/dt_{min}$), and maximum volume ($V_{max}$) at end-diastole ($N = 5$, biological replicates, ***$p < 0.001$ versus No treatment group, ##$p < 0.01$ and ###$p < 0.001$ versus No HEM-Ps group). **i** Quantitative graphs of the slope of end-systolic pressure-volume relationship (ESPVR) and the slope of the end-diastolic pressure-volume relationship (EDPVR) ($N = 4$ for No treatment and $N = 5$ for No HEM-Ps and HEM-Cf, biological replicates, **$p < 0.01$ versus No treatment group, #$p < 0.05$ versus No HEM-Ps group). Cardiac tissues prepared with hiPSC-derived RFP+ CMs, cardiac fibroblasts (CFs), and human umbilical vein endothelial cells (HUVECs) were cultured under each condition for 9 days and used for transplantation. Data are presented as means ± S.D. Statistical significance was determined using one-way ANOVA with Bonferroni's multiple comparison tests in (**c**, **d**, **e**, and **i**) and Tukey's multiple comparisons tests in (**g** and **h**).

difference as it was not subjected to ischemic/reperfusion injury[73] (Fig. 7e). Echocardiography measurements from 1 day before the induction of injury (normal state in all groups) showed that the rats in all groups initially had the same heart function (Supplementary Fig. 17).

When we performed pressure-volume (PV) loop analysis by inserting a catheter directly into the left ventricle (LV) 4 weeks after transplantation, the hemodynamic parameters of cardiac output and stroke volume were significantly higher in the HEM-Cf group (μF/HEM-CT) than in the other groups (Fig. 7f, g). Improvement was observed in the parameters of adverse cardiac remodeling, with a significant increase in the maximum rate of pressure change ($dP/dt_{max}$) and the minimum rate of pressure change ($dP/dt_{min}$), and a significant decrease in max volume ($V_{max}$) in the HEM-Cf group compared to the other groups (Fig. 7h). To assess the load-independent cardiac contractibility, we measured the change in PV loops after temporary occlusion of the inferior vena cava. In an injured heart, the slopes of the end-systolic pressure-volume relationship (ESPVR) and the end-diastolic pressure-volume relationship (EDPVR) become gentle and steep, respectively, due to impairment of the contraction and dilation functions[74]. When compared with the control groups (No treatment and No HEM-Ps groups), the HEM-Cf group showed a steeper slope of ESPVR and a gentler slope of EDPVR at 4 weeks post-transplantation (Fig. 7i and Supplementary Fig. 18). Taken together, the transplantation of cardiac tissues produced via 3D microfluidic HEM culture (μF/HEM-CT) consistently improved cardiac function and prevented adverse post-infarction remodeling.

Histological analysis was performed to examine the regenerative capacity of cardiac tissues 4 weeks after transplantation. MT staining of the harvested rat heart tissues showed that the fibrosis area of the LV wall was significantly smaller and the viable myocardium in the infarcted region was significantly larger in the HEM-Cf group (μF/HEM-CT) than in the control groups (Fig. 8a). Quantification of capillary density in the infarcted heart tissues after 4 weeks indicated that CD31+ capillary densities in the border and infarct zones were significantly higher in the HEM-Cf group than in the control groups (Fig. 8b). Similarly, the RFP-labeled cTnT+ CMs were distributed more abundantly over a larger area of the LV in the HEM-Cf group (Fig. 8c). To confirm the structural maturity of gap junctions, CX43 was stained. Lateralized CX43 indicates immature or injured gap junctions. The HEM-Cf group had less lateralized CX43 (yellow arrows) than did the No HEM-Ps group (Fig. 8d). Furthermore, in comparison to the control groups, the HEM-Cf group had more abundant viable CMs and significantly less denatured collagen stained with collagen hybridizing peptide (CHP) in the border and infarct zones (Fig. 8e, f). These findings suggest that our μF/HEM-CT generated using the integrated culture platform of 3D HEM hydrogel and dynamic microfluidic chip facilitates engraftment and retention of transplanted CMs, while significantly reducing the loss of host CMs and fibrosis in infarcted myocardium.

Overall, cardiac and vascular regeneration can be significantly improved by using μF/HEM-CT.

To provide the evidence of functional integration of CMs in the infarcted hearts, electrocardiogram (ECG) measurements were performed (Supplementary Fig. 19). ECG data were analyzed with a real-time heartbeat classification algorithm, generating waterfall plots for all detected heartbeat. In comparison to the No injury group (sham operation) and the HEM-Cf group, the No treatment group and the No HEM-Ps group showed typical pathological Q-waves and ST-segment elevation, indicative of myocardial infarction (Supplementary Fig. 19a). The HEM-Cf group showed a lower incidence of premature beats, and the PR, RR, QRS, and corrected QT (QTc) intervals recorded in the HEM-Cf group resembled those of the No injury group (Supplementary Fig. 19a, b). Additionally, the expression of CX43 in HEM-Cf group was well-organized and displayed connectivity with neighbor CMs up to 6 weeks (Fig. 8d and Supplementary Fig. 20). These electrophysiological and structural data suggest that the transplanted CMs successfully achieved functional integration within the host myocardium, mitigating the risk of arrhythmias. Overall, our in vivo results demonstrate the improved cardiac function, structural regeneration, and functional integration following the transplantation of cardiac tissues produced via 3D microfluidic HEM culture (μF/HEM-CT).

## Discussion
In this study, we advanced the development of human cardiac tissues utilizing a bioengineering platform that fully integrates three heart microenvironmental cues: extracellular (native heart-derived ECM), cellular (interactions with non-myocytes [ECs and CFs] in the heart), and dynamic cues (microfluidic device-based flow). The cardiac tissues generated in our culture platform provide macroscale (approximately 1.2 mm in diameter), highly mature, and functional models for a wide range of biomedical applications, including drug evaluation, disease modeling, and regenerative therapy.

The HEM developed in our study provided a highly reproducible and biocompatible 3D ECM microenvironment that mimics native heart tissue, allowing for the development and maturation of human cardiac tissue. We demonstrated that optimized decellularization procedures and consistently controlled donor conditions (e.g., age and gender) lead to the fabrication of HEM with minimal batch variability and safety issues (Fig. 1e, f and Supplementary Figs. 1–3). The residual DNA content in our porcine HEM and native porcine heart was $35.6 ± 12.4$ ng/mg and $4468.0 ± 97.1$ ng/mg, respectively, which means that 99.2% of DNA was removed (Fig. 1b). Notably, previous studies have set the guideline for competent decellularization as residual DNA content not exceeding 50 ng/mg[75–77]. Thus, our HEM meets this criterion for clinical application. Actually, a wide array of decellularized matrix products have been used for clinical applications. For instance, VentriGel (Ventrix, Inc.) derived from porcine decellularized myocardium was assessed in a Phase I clinical trial for the treatment of

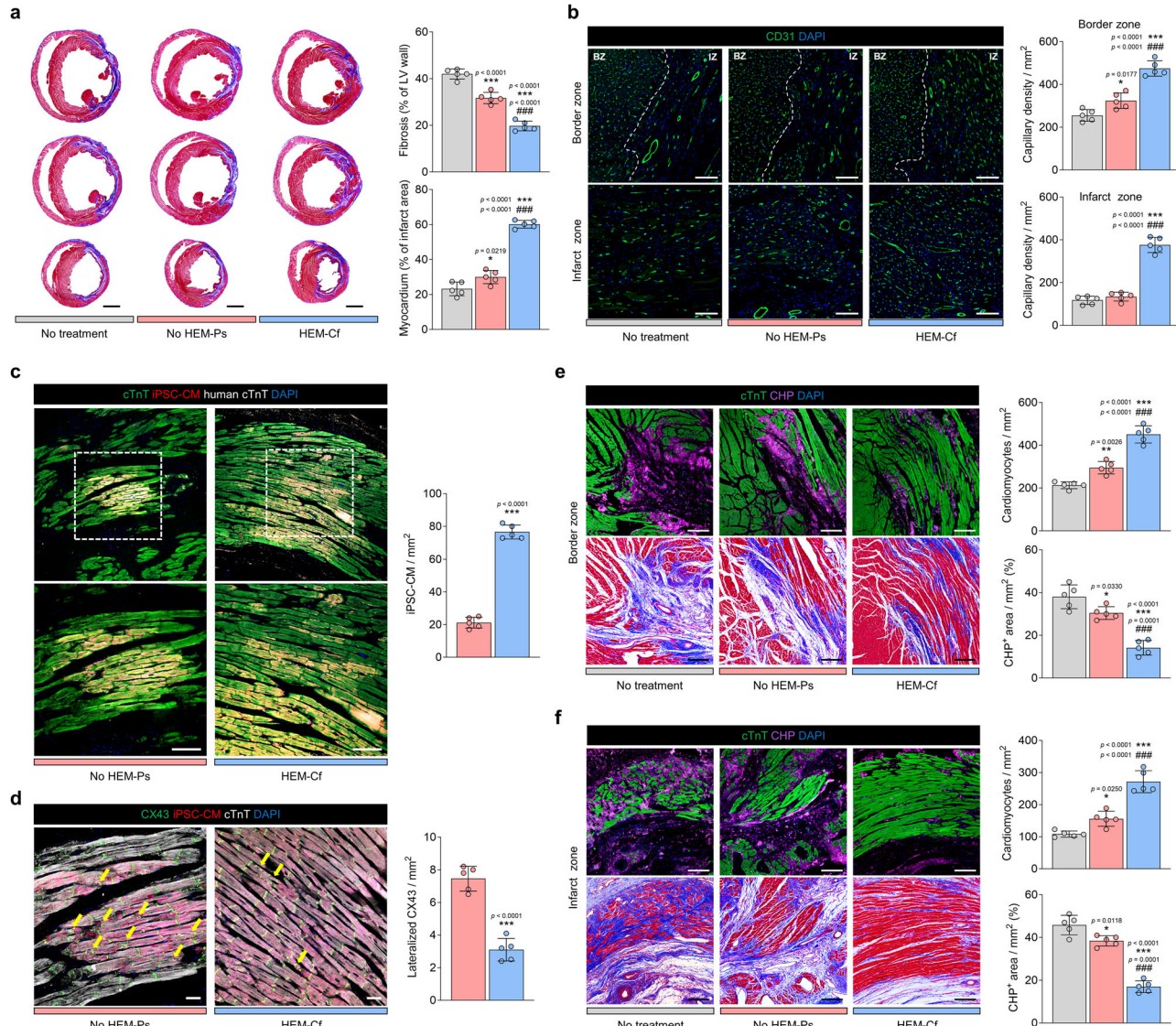

**Fig. 8 | Enhanced cardiac regeneration by human cardiac tissue transplantation in a rat model of myocardial infarction. a** Representative images of Masson's trichrome (MT) staining of ischemic hearts harvested 4 weeks after transplantation and quantification of the percentage ratio of fibrotic and viable myocardium (scale bars = 2 mm, N = 5, biological replicates, *p < 0.05 and ***p < 0.001 versus No treatment group, ###p < 0.001 versus No HEM-Ps group). **b** Representative images of capillary staining for CD31 (green) (scale bars = 200 μm) and quantification of capillary density in the infarct zone (IZ) and the border zone (BZ) 4 weeks post-transplantation (N = 5, biological replicates, *p < 0.05 and ***p < 0.001 versus No treatment group, ###p < 0.001 versus No HEM-Ps group). **c** Representative immunofluorescent images comparing retention rates of transplanted RFP+ cardiomyocytes (CMs) in each group (total CMs stained with cTnT [green] and human CMs stained with human cTnT [white] after 4 weeks; scale bars = 100 μm), and quantification of engrafted human induced pluripotent stem cell (hiPSC)-derived CMs in ischemic hearts (N = 5, biological replicates, ***p < 0.001). **d** Representative

immunofluorescent images of gap junctions expressed in transplanted RFP+ CMs (gap junctions stained with CX43 [green] and total CMs stained with cTnT [white]) 4 weeks post-transplantation. Yellow arrows indicate lateralized gap junctions (scale bars = 100 μm). The quantification graph shows the ratio of lateralized gap junctions to localized CX43 areas (N = 5, biological replicates, ***p < 0.001). Representative images of total CMs stained with cTnT (green) and denatured collagen stained with collagen hybridizing peptide (CHP, violet) on **e** the border zone and **f** the infarct zone after 4 weeks and their quantification data (scale bars = 100 μm, N = 5, biological replicates, *p < 0.05, **p < 0.01, and ***p < 0.001 versus No treatment group, ###p < 0.001 versus No HEM-Ps group). Cardiac tissues prepared with hiPSC-derived RFP+ CMs, cardiac fibroblasts (CFs), and human umbilical vein endothelial cells (HUVECs) were cultured under each condition for 9 days and used for transplantation. Data are presented as means ± S.D. Statistical significance was determined using unpaired two-sided Student's t-tests in (**c** and **d**) and one-way ANOVA with Tukey's multiple comparisons tests in (**a**, **b**, **e**, and **f**).

myocardial infarction[78]. Bovine decellularized vascular grafts, including Artegraft® and ProCol® were also commercialized for clinical use[79]. These present the safety and feasibility of decellularized tissue matrix as a therapeutic option for patients with cardiovascular diseases. Nonetheless, in-depth future studies would be required to ensure the immune tolerance of HEM by checking molecules associated with transplant rejection.

The three most abundant matrisome proteins present in HEM were fibrinogens (FGA, FGB, and FGG), which are widely used in

scaffolds to generate functional and mature CMs or to induce angiogenesis[80–82]. Three additional abundant HEM matrisome proteins included the three chains of collagen VI (COL6A1, COL6A2, and COL6A3). These proteins are essential ECM components for forming and regulating microfibril networks, which increase the integrity and stiffness of heart tissue[83–85]. The collagen VI subtypes, which also interact with cell membrane receptors and diverse components of ECMs, mediate the interactions between cells and surrounding ECMs, especially in ECs[86–88]. Based on the Human Protein Atlas[89], the HEM

contains 80–90 proteins that are specifically enriched in native heart tissues (Fig. 1g and Supplementary Fig. 2–4). Among the heart-enriched proteins, THBS4 (thrombospondin-4), LAMA2 (laminin α2 chain), and FBLN2 (fibulin-2) also contribute to cardiac maturation. THBS4 influences the function and remodeling of the myocardium and angiogenesis[90–92]. LAMA2 is primarily expressed in CFs[93,94] and can bind to CM receptors to promote CM maturation. FLBN2 regulates TGF-β signaling related to fibrosis progression, thus increasing the accuracy of our disease model for cardiac fibrosis[95]. In the HEM, the variety of heart-enriched matrisome and non-matrisome proteins and their networks not only promote differentiation and maturation of CMs and non-myocytes (ECs and CFs) but also orchestrate their complex interactions[96–98]. Overall, the HEM hydrogel is superior to conventional matrices or non-heart tissue-derived ECM hydrogels in improving the quality of fabricated cardiac tissues (Fig. 2).

Another key factor in our study is the incorporation of well-controlled perfusable flow, which addresses diffusional limitation issues and enables better nutrient and oxygen delivery to the developing cardiac tissues. Our microfluidic device serves as a miniature bioreactor, providing the cardiac tissues with a more sophisticated medium flow compared to conventional dynamic culture systems (e.g., spinner flasks and rotating bioreactors) in a relatively simple way[99–101]. Dynamic culture in our microfluidic chip significantly improved the viability and contractility of cardiac tissues compared to culture in 24-well plates with or without dynamic flow (Fig. 3d–i). To isolate the impact of dynamic culture in our microfluidic chip, we investigated the impact of dynamic flow in microfluidic chip and conventional well-plate under condition without HEM hydrogel by comparing the No HEM-Cf group with the No HEM-Pf group (Supplementary Fig. 21). We observed a significant upregulation in the expression levels of CM, EC, and CF markers in the No HEM-Cf group. These findings underscore the superiority of our microfluidic system over conventional methods for dynamic tissue culture, potentially leading to a higher level of myocardium regeneration.

Synchronized contractions were enhanced in cardiac tissues fabricated using the 3D microfluidic HEM culture (HEM-Cf group; μF/HEM-CT) (Fig. 3j and Supplementary Movie 2). In native heart tissue, synchronized contraction is electrically induced through gap junction-mediated cell-cell communication that occurs either directly between CMs or indirectly between CFs and CMs[102–104]. Thus, the dynamic flow in our chip system facilitates the maturation of the cells comprising the cardiac tissues and promotes cell-cell interactions for electrical synchronization by preventing apoptosis and promoting the formation of gap junctions. This is supported by the strong expression of gap junction marker CX43, which was present in both CMs and CFs in the HEM-Cf group on day 21 (Fig. 4i). *ACVRL1* is an EC marker, *MYH7* is a cardiac maturation marker, and *COL1A1*, *COL3A1*, and *FBN1* are markers of CFs and collagen fibril organization[105] that were among the genes specifically upregulated under the dynamic flow chip conditions compared with the static plate conditions (Fig. 4d). Moreover, *PDGFRB* induces proliferation of CMs and *SRPX2* modulates EC migration and tube formation[73,106]. *POSTN*, expressed in activated CFs, contributes to CM maturation and cardiac nerve development during the post-natal period[107,108]. In addition to cardiac transcriptomic signatures, the development and maturation of cardiac-specific ultrastructures can be promoted by microfluidic flow (Fig. 4j–l). Our microfluidic chip-based dynamic culture enables stable long-term maintenance of millimeter-scale tissue constructs, allowing for the production of a large quantity of mature cardiac tissue cells suitable for highly effective regenerative therapy (Figs. 7, 8).

In our study, we highlight the roles of HEM as a primary facilitator for enhancing the maturation of the constituent cells in cardiac tissues during in vitro culture. Interestingly, GO analysis revealed a significant upregulation of several genes upon the incorporation of HEM hydrogel (Supplementary Fig. 14c, d). Previous studies reported that overexpression of *Fosl1* enhanced the cardiac functions in a mouse myocardial infarction model[109], and Sfrp2 proteins promoted the expression of CM-specific genes, such as cardiac ion channels, in a myocardial infarction model[110]. These findings may suggest a potential mechanism behind the improved regenerative effects of our cardiac tissues by HEM incorporation. Actually, porcine decellularized heart-derived matrix alone, either in the form of an injectable hydrogel or patch, has shown therapeutic effects on myocardium regeneration[111,112]. Despite such potential therapeutic potential of HEM, myocardium regeneration in our study is more likely attributed to the transplanted cells rather than HEM. Prior to transplantation, we induced the gentle fragmentation of cardiac tissues using collagenase for reliable injection[24,113,114]. During this process, the majority of collagen components are removed, leaving only trace amounts of other residual ECM constituents in cardiac tissues. Therefore, we speculate that the HEM had limited direct effects on tissue regeneration in vivo. This underscores the importance of the cellular condition of cardiac tissues in influencing the level of myocardium regeneration.

We have demonstrated the advantages of our cardiac models over previously reported cardiac models, such as EHT[6,115], heart-on-a-chip[116–118], and cardiac organoids[119–121]. EHT can be manufactured on a relatively large scale in various shapes based on the mold structure, but it has a high risk of necrosis due to the absence of flow[23]. Heart-on-a-chip is based on a microfluidic system that enables dynamic culture, but it often requires additional equipment, including a syringe pump and tubing, which limits the simultaneous application of a large number of chips[122,123]. In contrast to these previous models, our cardiac tissue model utilizes a simple and user-friendly dynamic system without specific molds or additional equipment. Mass cultivation of mature cardiac tissues can be achieved by simply stacking the tube-free chips on axial rockers. For example, 250 chips (approximately 20 grams per chip with three chambers) could be housed and incubated simultaneously using an axial rocker with a permissible rocking weight of 5 kg, allowing for animal study on about 75 rats. In addition, if the aim is to produce a greater quantity of cardiac tissues for large animal studies, it would be entirely feasible to fabricate the chips with adapted design that enables the cultivation of an increased amount of cardiac tissue by expanding the number of chambers per chip (48 - 96 chamber)[124]. Cardiac organoids can recapitulate complex configurations, including cardiac chambers, during embryonic heart development[119–121]. While they are suitable for developmental studies owing to their fetal heart tissue-like features[23], they may not be the best option for disease modeling and drug screening. In contrast, our cardiac tissues composed of differentiated heart-constituting cells achieve high maturity and functionality through dynamic microenvironments that mimic the heart. Our human cardiac model can significantly improve and enhance the reliability of drug testing and disease modeling. Our models demonstrated improved sensitivity to drugs and accurately distinguished electrophysiological alterations and impairments caused by various drugs at different levels of cardiotoxicity (Fig. 5f). Additionally, our tissue models were able to recapitulate cardiac diseases caused by congenital genetic defects or fibrotic signaling (Fig. 6).

Our cardiac tissues employ a spheroid-type model for simple and accessible fabrication that can be readily replicated by anyone. However, our approach encounters inherent limitations from the isotropic spheroid format, such as maturation deficiency. To address this, we employed a dual-boosting approach as aforementioned. First, pre-gel solution of HEM hydrogel enriched with heart-specific ECM was administered at the onset of 3D spheroid formation. Consequently, all the constituent cells forming the cardiac tissue could directly engage with the heart-specific ECM for promoting maturation. Second, we harnessed the dynamic flow generated by a microfluidic device. We found that CMs were more prevalent at the border zone (the region of the outer circle occupying over half the diameter) of our multicellular cardiac spheroid models due to the compartmentalization of three cell

types (Fig. 4h). Upon exposure to dynamic flow, ECs spontaneously localized to the interior and CFs tended to be positioned in the exterior, a phenomenon previously reported in other studies[68,125]. This suggests that our cardiac tissue model with dynamic flow could mimic the native heart development process, as validated in heart-forming organoids[119]. Although many CMs were found at the border zone in our cardiac tissue, a mature thick CM layer was formed (Fig. 4i). This CM layer exhibited enhanced contractility and synchronized beating (Fig. 3g–i and Supplementary Movies 1, 2), making it useful for cardiotoxicity testing.

The spherical shape of our cardiac tissues poses a limitation in directly measuring the force of contraction. Thus, we used MUSCLE-MOTION software to measure the amplitude of contraction (Figs. 3g–i, 6k, o). This software is extensively being utilized in 3D cardiac spheroid research, and the contraction amplitude derived from measurement of EHT with this software exhibited a strong linear correlation with force of contraction[126]. In addition, several studies have highlighted the effectiveness of using contraction amplitude for detecting drug-induced contractile responses and contractile changes resulting from myocardial infarction[17,127]. BPM of cardiac tissues was also analyzed using this software (Fig. 3i). hiPSC-derived CMs are known to beat spontaneously due to their resemblance to fetal properties, while adult ventricular CMs are electrically quiescent in the absence of external stimulation[128]. The human fetal heart typically beats between 120–160 BPM, gradually decreasing to 60–100 BPM during maturation to the adult heart[129]. In our cardiac tissues of the HEM-CF group, the BPM falls within the range of 60–80, which is close to the level of normal adult heart beats[38]. This suggests that although our hiPSC-derived cardiac tissues may not reach the level of maturity seen in the adult heart, the CMs of our cardiac tissues exhibit a noteworthy advancement in the maturity, compared with typical hiPSC-derived CMs that are considered highly immature.

Nonetheless, we need to further improve our cardiac tissue model to achieve the maturity seen in the adult heart. Other engineering methods, such as cardiac alignment and electrical/mechanical stimulation, have been widely employed in advancing cardiac tissue maturation[6,9,130]. Notably, the utilization of decellularized tissue matrix provides distinct advantages over other methods, including a relatively short period for maturation and the ability to offer phenotype guidance even before cardiac specification[131,132]. Throughout in vivo developmental processes, the combination of ECM interaction, non-myocyte interaction, and electrical/mechanical stimulation synergistically contributes to cardiac maturation[133]. This highlights the necessity for engineering approaches that integrate these elements concurrently into our cardiac tissues, resulting in synergistic impact of biochemical, electrical, and mechanical cues[134]. Since electrical/mechanical stimulation techniques are typically applied during the culture period, they can be readily integrated with HEM hydrogel culture for further promoting maturation of cardiac tissues. This combination approach would be worth being validated in the future studies. Furthermore, it is necessary to fabricate cardiac organoids with void chambers whose volume changes with cardiac contraction and investigate the fluid flow and oxygen diffusion in such more realistic heart models. In our current study, any form of voids in the cardiac tissues has not yet been detected. This might be improved through the development of cardiac tissues containing endocardial chambers through the incorporation of endocardial cells[121].

In summary, we validated the versatility of human cardiac tissues fabricated through our microfluidic HEM culture system for three major biomedical applications: drug evaluation, disease modeling, and transplantation. We anticipate that our versatile cardiac tissues will be established as practically available human in vitro cardiac models and provide therapeutic regimens for in vivo cardiac regeneration through further translational research.

## Methods

### Fabrication of HEM

Fresh porcine heart tissues of 6-month-old adult pigs were purchased from the local stock market (Majang-dong, Seoul, Korea). Female pigs were utilized because they are overwhelmingly more common in the marketplace. Decellularization of the porcine heart was performed with only a minor change to our reported protocol[31]. In detail, left ventricles were used because they are the largest parts in the heart. Prior to decellularization, heart tissues were chopped into small pieces (approximately 0.5-mm cubes) to increase the surface-to-volume ratio and facilitate the decellularization process. The chopped heart tissues were washed vigorously with distilled water for 6 h to drain the remaining blood and contaminants. To enhance the efficiency of decellularization process, all tissues were treated with solution at a volume ratio of 1:10. Washed heart tissues were incubated in the following solutions: 24 h in 1% sodium dodecyl sulfate (SDS, Thermo Fisher Scientific, Waltham, MA, USA) with a change of the solution every 12 h, 6 h in 1% (v/v) Triton X-100 (Sigma-Aldrich, St. Louis, MO, USA) and 0.1% (v/v) ammonium hydroxide (Sigma-Aldrich), and 16 h in distilled water with a change of water every 6 h to remove remaining bubbles from use of detergents. Decellularization of fresh porcine stomach, small intestine, and skeletal muscle was performed as described in our reported protocol[24,135]. In detail, porcine stomach and skeletal muscle tissues were fragmented into small pieces, whereas small intestine tissues were cut longitudinally and sliced into long pieces (approximately 10 cm). Stomach and small intestine tissues were incubated in 1% (v/v) Triton X-100 and 0.1% (v/v) ammonium hydroxide for 48 h. The solution was replaced at 6 h, 12 h, 24 h, and 36 h. Subsequently, the tissues were thoroughly rinsed with distilled water for 16 h, with refreshed every 6 h to ensure optimal cleansing. Skeletal muscle tissues were washed in a series of solutions: 0.2% (v/v) trypsin-ethylenediaminetetraacetic acid (EDTA, Thermo Fisher Scientific) for 3 h, 0.5% (v/v) Triton X-100 for 12 h, and a mixture of 1% (v/v) Triton X-100 and 0.2% (w/v) sodium deoxycholate (Sigma-Aldrich) for 22 h. Then, the tissues were rinsed with distilled water for 16 h, with the water exchanged every 6 h. Sterilization was conducted by washing the decellularized tissues with 1% (v/v) penicillin-streptomycin (Thermo Fisher Scientific) for 2 h and multiple times with sterile distilled water. The entire decellularization process was performed under constant 180 rpm rotation at 4°C. The decellularized tissue matrices were lyophilized and stored at 4°C until further use.

Lyophilized HEM and other tissue matrices were digested with 8 mg/ml (w/v) pepsin (Sigma-Aldrich) in 0.02 M HCl to a final concentration of 10 mg/ml (w/v) solution. To prepare pre-gel solutions of tissue-derived ECM hydrogel, 10% (v/v) 10× phosphate-buffered saline (PBS, Sigma-Aldrich) and distilled water were added to the ECM solution and the pH was adjusted to approximately 7.4 with 0.5 M sodium hydroxide (Sigma-Aldrich).

### Fabrication of the microfluidic chip

The device consisted of three patterned PDMS layers with microfluidic channels connecting three culture chambers and two medium chambers, along with a non-patterned PDMS film at the bottom of the device as seen in Fig. 3a. Microfluidic patterned layers were prepared using a soft lithography procedure[34,124,136]. The 300-μm-thick patterned master wafer (Amed Inc., Seoul, Korea) was fabricated using photolithography. The PDMS solution was prepared from a mixture of SYLGARD™ 184 Silicone Elastomer Base (Dow Silicones Corporation, Midland, MI, USA) and SYLGARD™ 184 Silicone Elastomer Curing Agent (Dow Silicones Corporation) at a 10:1 ratio (w/w). The PDMS solution was poured on the wafer and cured in a dry oven at 60°C for 2 h after degassing with a vacuum desiccator. The PDMS layers were cut out of the wafer and holes were made for culture and medium chambers using various punch sizes. Three patterned PDMS layers were assembled immediately after surface treatment with oxygen

plasma (CUTE; Femto Science, Seoul, Korea). The assembled devices were transferred to a dry oven at 60°C for 4 h for tight bonding. Prior to experimentation, the fabricated microfluidic chips were sterilized in an autoclave and exposed to ultraviolet light.

## Generation of cardiac tissues

This study using hiPSCs was approved by the Institutional Review Board (IRB) of Yonsei University (Permit Number: 7001988-202004-BR-844-01E). To fabricate human cardiac tissues, CMs, CFs, and ECs were detached for 3D co-culture. hiPSC-derived CMs were treated with 290 U/ml collagenase type 2 (Thermo Fisher Scientific) for 1 h and sequentially treated with 0.05% Trypsin/EDTA (Thermo Fisher Scientific) for 5 min. hiPSC-derived CFs and ECs (human umbilical vein endothelial cells; HUVECs, Lonza, Basel, Switzerland) were dissociated using 0.05% trypsin/EDTA for 5 min. CM:CF:EC were mixed at a 2:1:1 ratio and centrifuged at $200\,g$ for 5 min. The mixed cell ratio was determined based on several references reporting cardiac tissue production with these three cell types[137–139]. The cells were encapsulated at $2.0 \times 10^5$ mixed cells per 20 μl of ECM hydrogel and incubated at 37°C for 30 min. For the No ECM group, $2.0 \times 10^5$ mixed cells were seeded into each well of 96-well U-bottom plates (Corning, Corning, NY, USA) and centrifuged at $110\,g$ for 6 min.

To compare culture matrices, decellularized ECM hydrogels (HEM and other tissue-derived ECMs), collagen gel, fibrin gel, GFR-Matrigel (Corning), and hESC-qualified Matrigel (ESC-Matrigel, Corning) were evaluated for cell encapsulation and cardiac tissue formation. For collagen gel, rat tail collagen type 1 (Corning) was mixed with 10× PBS, distilled water, and 0.5 M sodium hydroxide to a final concentration of 2 mg/ml, and the pH was adjusted to approximately 7.4. The pre-gel solution of fibrin gel was prepared with 100 μl fibrinogen (10 mg/ml, Sigma-Aldrich) and 1.6 μl thrombin (50 U/ml, Sigma-Aldrich) and adjusted to a final concentration of 2 mg/ml.

Prior to the culture of cardiac tissues in microfluidic chips, the HEM group was cultured in 24-well culture plates and the No HEM group was cultured in 96-well U-bottom plates on the first day of fabrication for initial stabilization. The formed cardiac tissues were transferred to microfluidic chips (Chip flow (Cf) groups: No HEM-Cf and HEM-Cf) and cultured on an axial rocker (4 rpm, CRS-350; Lab Companion, Daejeon, Korea) on the second day. The plate flow (Pf) group (HEM-Pf) was cultured in 24-well culture plates at 4 rpm and the plate static (Ps) groups (No HEM-Ps and HEM-Ps) were cultured in a static state.

The culture medium for cardiac tissues was a mixture of RPMI/B27 medium (Thermo Fisher Scientific) and Endothelial Cell Growth Medium-2 (EGM, Lonza) at a ratio of 1:1. Medium was changed every other day. All cardiac tissues were cultured in 5% $CO_2$ at 37°C. Various analyzes including histology, immunostaining, quantitative PCR (qPCR), and RNA sequencing were performed on days 7, 14, or 21 depending on the experiments. The projected area of cardiac tissues was quantified using ImageJ software (National Institutes of Health, Bethesda, MD, USA).

## Cardiac disease modeling

For LQTS modeling, the GM25305 hiPSC line (LQT2 hiPSC) obtained from the Coriell Institute for Medical Research (Camden, NJ, USA) was used. LQT2 patient iPSC-derived CMs (LQT2-CMs) and LQT2 patient iPSC-derived CFs (LQT2-CFs) were differentiated using the same protocols for normal CM and CF, as described in the Supplementary Information[140,141]. LQT2-CMs, LQT2-CFs, and HUVECs were encapsulated at a 2:1:1 ratio in 2 mg/ml HEM hydrogel in 24-well plates. On day 1 of tissue fabrication, formed LQTS cardiac tissues were transferred to microfluidic chips and cultured on an axial rocker in 5% $CO_2$ at 37°C until day 21.

For cardiac fibrosis modeling, normal hiPSC-derived CMs, normal hiPSC-derived CFs, and HUVECs were mixed at a 2:1:1 ratio in 2 mg/ml HEM hydrogel in 24-well plates and transferred to microfluidic chips as above. For fibrosis induction, 50 ng/ml TGF-β1 (PeproTech, Cranbury, NJ, USA) was added to the cardiac tissues every other day from day 1 to day 21 of the culture. For anti-fibrotic drug testing, the fibrotic cardiac tissues were treated with 5 μM losartan (Sigma-Aldrich) on day 14 and cultured for an additional 7 days.

## Myocardial infarction induction and cardiac tissue transplantation

Animal studies for transplantation were approved by the Institutional Animal Care and Use Committee (IACUC) of the Catholic University of Korea (approval number: CUMC-2021-0135-05). All rats were maintained in the housing condition with a temperature of $21 \pm 2\,°C$, a humidity of $50 \pm 10\%$, ventilation of 10–15/h, light of 150–300 Lux, and noise of less than 60 dB. All procedures conformed to NIH guidelines or the guidelines issued by Directive 2010/63/EU of the European Parliament for the protection of animals used in scientific research. Fischer 344 rats (F344/NHsd, male, 8 weeks old, 160–180 g; KOATECH, Pyeongtaek, Korea) were anesthetized using 2% isoflurane (Hana Pharm, Seongnam, Korea) and intubated using an 18-gauge intravenous catheter (Korea Vaccine, Seoul, Korea). Only male rats were employed to minimize variation in the outcomes of ischemia/reperfusion injury, considering biological sex[142]. Rats were mechanically ventilated, and a left thoracotomy was performed after the chest wall was shaved.

To induce myocardial ischemia, the LAD artery was ligated for 1 h using a 7–0 prolene suture (Ethicon, Inc., Somerville, NJ, USA). After 1 h of ischemia induction, the knot was removed to allow for reperfusion of the LAD. The chest was aseptically closed and disinfected after surgery. One week after ischemia/reperfusion injury, base echocardiography (Affiniti 50 G, Philips, Andover, MA, USA) was performed prior to transplantation of cardiac tissues. After that, the left thoracotomy was re-performed, and the prepared cardiac tissues were transplanted at two different border zone sites in the infarcted heart.

For transplantation, cardiac tissues harvested 9 days after fabrication were fragmented with collagenase type 2 (Thermo Fisher Scientific). The rats were re-anesthetized for transplantation using isoflurane inhalation, intubated, and mechanically ventilated. The chests of the animals were reopened, and the infarct area with collagen deposition was inspected. Subsequently, we injected cardiac tissues at two spots in the border zone of infarcted myocardium. Ten fragmented cardiac tissues containing $1.0 \times 10^6$ RFP-CMs were suspended in 50 μl of PBS for injection into one rat. The injection dose of cardiac tissues was standardized based on the quantity of CMs, which was identical to the number of CMs transplanted in the previous study[143]. The No treatment group was injected with PBS only. All animals received daily immunosuppressants composed of azathioprine (2 mg/kg, Korea United Pharm Inc., Seoul, Korea), cyclosporine A (5 mg/kg, CKD Pharm, Seoul, Korea), and methylprednisolone (5 mg/kg, Hanlim Pharm, Seoul, Korea) as described in previous studies[144,145].

For evaluating the therapeutic efficacy of cardiac tissue transplantation, serial echocardiography was performed 1, 2, 4, and 6 weeks after transplantation, and hemodynamic measurements were taken at 4 weeks. Histological analyzes and immunostaining of rat heart tissue sections were also conducted 4 weeks after transplantation.

## Statistical analysis and reproducibility

Statistical analyzes were conducted using GraphPad Prism 10 (GraphPad, La Jolla, CA, USA). Statistical significance was determined using unpaired, two-sided Student's t-tests, one-way analysis of variance (ANOVA) with Tukey's multiple comparisons tests, one-way ANOVA with Bonferroni's multiple comparisons tests, and one-way ANOVA with Dunnett's multiple comparisons tests based on the test requirements. All details regarding experimental replications, biological replicates (N), and statistical tests are included in the corresponding figure legends.

**Reporting summary**

Further information on research design is available in the Nature Portfolio Reporting Summary linked to this article.

## Data availability

All datasets supporting this study are available within this article and its supplementary information file. Proteomics data are available at the ProteomeXchange Consortium via the PRIDE partner repository with the dataset identifier "PXD042077" [proteomecentral.proteomex-change.org/cgi/GetDataset?ID=PXD042077]. Proteomics data from LC-MS/MS were processed using the UniProt database [uniprot.org]. Matrisome proteins were identified and categorized using the Matrisome Project database [sites.google.com/uic.edu/matrisome/]. Heart-enriched proteins were identified using the Human Protein Atlas [proteinatlas.org]. RNA-sequencing data have been deposited to the Gene Expression Omnibus (GEO) public repository under accession codes "GSE231493" [ncbi.nlm.nih.gov/geo/query/acc.cgi?acc=GSE231493]. All raw data from quantitative analyzes in our study are provided as a single Excel file 'Source Data'. All microscopic images and other data generated for this study are available from the corresponding author on reasonable request, because they are too large and complicated to be deposited. Source data are provided with this paper.

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

## Acknowledgements

This work was supported by the National Research Foundation of Korea (NRF) grants (2021R1A2C3004262 to S.-W.C. and 2022R1A2C2009067 to H.-J.P.) funded by the Korea government, the Ministry of Science and ICT (MSIT). This work was supported by the Bio & Medical Technology Development Program (2022M3A9B6082675 to S.-W.C.) of the NRF funded by the MSIT and the Technology Innovation Program (20024298 to S.-W.C., Materials/Components Technology Development Program) funded by the Ministry of Trade, Industry & Energy (MOTIE, Korea). This work was also supported by the Institute for Basic Science (IBS-R026-D1) and the Yonsei Signature Research Cluster Program (2023-22-0012 to S.-W.C.). This work was supported by the Yonsei Fellow Program funded by Lee Youn Jae.

## Author contributions

S.-W.C. and H.-J.P. conceived and supervised the study. S.M. and S.K. designed and performed most of the experiments and data analysis. W.-S.S. conducted the experiments on cardiac tissue transplantation and performed the related analyzes. H.K., S.B.H. and Y.-J.C. conducted the animal experiments. Y.S.C. and M.J.L. performed the tissue decellularization. Y.S.C., M.J.L. and S.-H.J. performed the proteomic analysis. H.J. and B.C. fabricated the microfluidic devices and conducted the computational simulation analyzes. S.-J.L. and H.-A.L. performed the cardiac tissue-based drug evaluation experiments. I.J. and J.-U.P. assessed the electrophysiological characteristics of cardiac disease models. J.-H.P., J.-J.K., K.B. and Y.-G.K. supervised the analysis and assisted with data interpretation. S.-W.C., H.-J.P., S.M., S.K. and W.-S.S. wrote the manuscript with input from all authors.

## Competing interests

S.-W.C., S.M., Y.S.C. and S.K. are co-inventors on a patent application (Korean Patent 10-2021-0020145, pending) related to decellularized heart-derived ECM for cardiac tissue generation and culture. S.-W.C. is a chief technology officer (CTO) of Cellartgen, Inc., Republic of Korea. The remaining authors declare no competing interests.

## Additional information

[1]Department of Biotechnology, Yonsei University, Seoul 03722, Republic of Korea. [2]Cellartgen, Seoul 03722, Republic of Korea. [3]Department of Biomedicine & Health Sciences, College of Medicine, The Catholic University of Korea, Seoul 06591, Republic of Korea. [4]Division of Cardiology, Department of Internal Medicine, Seoul St. Mary's Hospital, The Catholic University of Korea, Seoul 06591, Republic of Korea. [5]Department of Predictive Toxicology, Korea Institute of Toxicology, Daejeon 34114, Republic of Korea. [6]Department of Materials Science and Engineering, Yonsei University, Seoul 03722, Republic of Korea. [7]Department of Chemical Engineering, Soongsil University, Seoul 06978, Republic of Korea. [8]Department of Thoracic and Cardiovascular Surgery, Seoul St. Mary's Hospital, College of Medicine, The Catholic University of Korea, Seoul 06591, Republic of Korea. [9]Division of Cardiology, Department of Internal Medicine, Eunpyeong St. Mary's Hospital, College of Medicine, The Catholic University of Korea, Seoul 03312, Republic of Korea. [10]Department of Biomedical Sciences, City University of Hong Kong, Kowloon 999077, Hong Kong. [11]Department of Neurosurgery, Yonsei University College of Medicine, Seoul 03722, Republic of Korea. [12]Center for Nanomedicine, Institute for Basic Science (IBS), Seoul 03722, Republic of Korea. [13]Graduate Program of Nano Biomedical Engineering (NanoBME), Advanced Science Institute, Yonsei University, Seoul 03722, Republic of Korea. [14]Cell Death Disease Research Center, College of Medicine, The Catholic University of Korea, Seoul 06591, Republic of Korea. [15]These authors contributed equally: Sungjin Min, Suran Kim, Woo-Sup Sim. ✉e-mail: cardioman@catholic.ac.kr; seungwoocho@yonsei.ac.kr

