## [Peer Review File · Nature Communications]

REVIEWER COMMENTS

Reviewer #1 (Remarks to the Author):

This study by Min et al. discusses the development of a culture system for producing engineered functional cardiac tissue spheroids from pluripotent stem cells derived cardiomyocytes, primary fibroblasts, HUVEC and decellularized heart matrix. This system uses a microfluidic chip and bidirectional perfusion to create a dynamic culture environment to support the engineered cardiac spheroids. The combined use of decellularized heart matrix and dynamic perfusion allowed the cardiac tissues to mature structurally and functionally. These human cardiac tissue spheroids have shown improved efficacy in several areas. They have been used successfully for drug testing, particularly in assessing the arrhythmia risk of various small molecules. They have been used for modeling diseases like Long QT Syndrome and cardiac fibrosis, and also have shown promise for regenerative therapies, such as treating myocardial infarction. This study spans across many research areas from bioengineering to drug testing to regenerative medicine and is extremely comprehensive and well presented. With that said, the biggest limitation with this study is the lack of cardiomyocyte alignment because of the isotropic tissue spheroid format. In addition, it seems cardiomyocytes are present mostly near the surface of the spheroid where mechanical tension is the strongest. This is not surprising, but it indicates that the spheroid format is not an efficient way of maturing cardiomyocytes because only the surface of the spheroid presents a desirable environment for cardiac maturation. Nonetheless, because of the extent of characterization and demonstration of applications in this paper, this study is still highly valuable to the community. I provided the following comments to help improve the manuscript.

The study is comprehensive, but it lacks a bit of understanding on the mechanism. For example, are HEM really contributing to myocardium regeneration? How much HEM are still present in the spheroids by the time the spheroids are implanted? Are all the benefits coming from having more healthy and matured cardiomyocytes in the spheroids? I think this is important to know considering the most novel aspect of this paper is in the use of HEM.

The significant improvement of cardiac function by the HEM-Cf group compared to the no HEM-Ps group is great. But it's unclear if this improvement is caused by HEM or perfusion on tissue spheroids. The improvement of perfusion on tissue spheroids is not a new finding and has been done before. In fact, I wonder if cardiac spheroids without any ECM cultured in a stirred tank bioreactor or a dish under constant rocking will provide the same level of regeneration.

The histology sectioning of fibrotic regions should be quantified. This increase in production of collagen is a major hallmark of fibrosis. This data should be quantified by experimental replicates.

To verify the model is effective, can an anti-fibrotic drug be used to show the reversal of the fibrosis progression?

It's unclear how many spheroids can fit inside the perfusion chip. For large animal studies in the future, large quantities of spheroids will be needed. How scalable is this platform?

The Author clearly demonstrated a good level of maturation of these tissues, but it's unclear how the maturation level of these tissue spheroids compare with other engineered cardiac tissue that can induce cardiac alignment and actively maturing the tissues with electrical and/or mechanical stimulation. For regenerative medicine application, using decellularized matrix from porcine will always present some risks from residual DNA fragments if the matrix is not decellularized completely. Therefore, it's unclear how much benefit to tissue maturation does decellularized matrix provide when compared with maturation induced from electrical and/or mechanical stimulation and if this benefit justify the risk. This should be discussed.

The platform doesn't allow the measurement of force of contraction (only demonstrated the amplitude of contraction) which is an important readout for cardiac drug testing. This should be discussed.

The title of the paper needs some improvement. Microfluidic heart extracellular matrix doesn't make a lot of sense. I don't think the microfluidic component is so significant that it needs to be highlighted in the title. Using this type of microfluidic device to apply perfusion to spheroids is a very common approach. The focus should be on the heart extracellular matrix.

Reviewer #2 (Remarks to the Author):

This is a well-designed experiment, in my opinion, there are still some flaws in this manuscript :

1. The author chose three types of cardiovascular lineage cells (CMs, ECs, and CFs). Why were other cell types such as vascular smooth muscle cells not considered? How are the ratios and cell numbers of these three types of cells determined?
2. About the porcine's heart. How old is the donor porcine? Porcine HEM samples derived from ventricular or atrial tissue?
3. The immune rejection of porcine HEM is very important for its possible subsequent clinical application. Digested ECM is also highly immunogenic. Is it reasonable for the author to discuss activated

fibroblasts, leukocytes, M1 macrophages and M2 macrophages with only four molecules of α -SMA, CD11b, iNOS and CD206?

4. Can the simulated oxygen content truly reflect the oxygen content inside the heart tissue? Is there a way to directly or indirectly monitor oxygen levels? An example is the consumption rate of dissolved oxygen in the medium.
5. The article shows that HEM shows improved efficacy in drug testing and disease modeling. Other methods should be used to prove the improvement of efficacy, such as analyzing the possible mechanism of improvement through transcriptomic or proteomic methods, and verifying the related molecules of improved efficacy through PCR and WB.
6. Given the potential immunogenicity of HEM, is it necessary to evaluate its therapeutic potential in a rat model of myocardial infarction? To assess the therapeutic potential of HEM, transplant rejection should be considered, for example by detecting molecules associated with transplant rejection.
7. In Figures 2e and 2k, which part of the experiment did the statistical data of c-TnT come from?
8. In Figure 3e and Figure 4f, the inner layer cells of the HEM-Pf and HEM-Ps groups partially express CD31, but the number of sensory cells is quite large, are they all endothelial cells?
9. #1, #2 and #3 in Figure 4c are easily mistaken for 3 repetitions. suggested modification
10. Figure 6a, the fabrication process of IPS in this schematic diagram is not the work of this manuscript

Reviewer #3 (Remarks to the Author):

The manuscript by Min et al describes the development of spherical cardiac tissues by co-culture from iPSC derived cardiomyocytes and fibroblasts using a simple microfluidic device that provides gravity driven flow. The tissues are based on a hydrogel from decellularized porcine hearts. The work provides thorough characterization by a range of methods as well as the use of the tissue for drug testing, disease modelling and in vivo therapy. Compared to other published methods, this system is easy to implement and could find use because of those reasons.

- The tissues do not exhibit a well developed cardiac syncytium, from TnT staining. The tissues are rather immature compared to other methods.
- Different levels of cell-gel compaction with control hydrogels vs decellularized ECM suggest there might be different mechanical properties at play that could contribute to the outcome
- Comparison of decellularized ECM from different organs is elegant
- For ventricular myocardium, quiescent tissues are better, ie a sign of improved maturation. High spontaneous beating of ventricular tissues is not necessarily a plus.

- Sarcomeres are not well developed compared to other approaches.
- More convincing evidence is needed for the presence of T-tubules
- The hallmark of LongQT syndrome is prolonged action potential. It should be shown if one is to claim disease modelling of LongQT. Additionally, all the same contraction duration with Long QT is shown. One would expect prolongation, but it may not be necessary, so long Aps are prolonged.
- Mason's Trichrome evidence of fibrosis is not convincing. Collagen is appears very clearly blue in Mason's trichrome. More convincing evidence of fibrosis is needed.
- Strong functional data on transplantation are strong in terms that both echo and hemodynamics measurements were provided. But the experiment ends at 4 weeks, it should be taken to 6 weeks, that is the time when pathological remodeling is over after MI. Often treatments may fail at 6 weeks.
- Details and dosing of tissue injection are needed.
- The initial injury was mild, often people use larger infarcts, so that Ejection fraction goes down to 30%. Mild injury is easier to reverse.
- Provide measurements for ventricular wall thickness.
- Evidence of integration is weak, there is no optical mapping or any other evidence of functional integration of the graft.
- Please check the text in the methods on endothelial cell differentiation. Where do ECs in the tissues come from?
- Validate assumptions for the use of Brinkman equation for tissue modelling. If flow is assumed through the hydrogel/tissue of course modelling will give the higher levels of oxygen. However, compact and dense tissues such as the heart tissue in vivo do not have flow (convection) in the tissue space. Flow happens through the capillary bed. The use of this equation then assumes cell density is inherently low, which is opposite from the main claim in this paper, that this is a dense cardiac tissue.
- There could be some flow through the voids of the tissue due to the contraction (that would almost act like a pumping motion), but this all has to be avlaidated. Brinkman equation is for porous media, is this really porous media if cells are there and connected to one another as the authors claim.
- Along those lines, please provide cell density across the diameter of the tissue, as well as viable or apoptotic cell density to prove the claim cell density was uniform.
- Critical details are missing in the methods about how exactly the tissue was formed, about the MI model and details about the tissue injection into the heart are also missing.
- It would be really nice to show movies of contraction and Ca²⁺ transients.

RESPONSE TO REVIEWER #1

General comments: This study by Min et al. discusses the development of a culture system for producing engineered functional cardiac tissue spheroids from pluripotent stem cells derived cardiomyocytes, primary fibroblasts, HUVEC and decellularized heart matrix. This system uses a microfluidic chip and bidirectional perfusion to create a dynamic culture environment to support the engineered cardiac spheroids. The combined use of decellularized heart matrix and dynamic perfusion allowed the cardiac tissues to mature structurally and functionally. These human cardiac tissue spheroids have shown improved efficacy in several areas. They have been used successfully for drug testing, particularly in assessing the arrhythmia risk of various small molecules. They have been used for modeling diseases like Long QT Syndrome and cardiac fibrosis, and also have shown promise for regenerative therapies, such as treating myocardial infarction. This study span across many research areas from bioengineering to drug testing to regenerative medicine is extremely comprehensive and well presented. With that said, the biggest limitation with this study is the lack of cardiomyocyte alignment because of the isotropic tissue spheroid format. In addition it seems cardiomyocytes are present mostly near the surface of the spheroid where mechanical tension is the strongest. This is not surprising, but it indicates that the spheroid format is not an efficient way of maturing cardiomyocytes because only the surface of the spheroid present a desirable environment for cardiac maturation. Nonetheless, because of the extent of characterization and demonstration of applications in this paper, this study is still highly valuable to the community. I provided the following comments to help improve the manuscript.

Response: We sincerely appreciate the reviewer for your insightful and constructive comments on our manuscript. They have been extremely valuable and helpful in improving the quality of our manuscript. We have thoroughly revised our manuscript based on the comments of the reviewer, and we hope that this revised version of manuscript solves all issues raised by the reviewer and meets your expectation for publication. The revised parts in the manuscript have been highlighted in red.

The cardiac tissues that we reported in this study employ a spheroid-type model for simple and accessible fabrication that can be readily replicated by anyone. However, as

pointed out by the reviewer, our approach encounters inherent limitations from the isotropic spheroid format. To create a tissue model that balances simplicity in fabrication with advanced cardiac features, we employed a dual-boosting approach to tackle the maturation deficiency in a spheroid-type cardiac tissue.

First, we utilized HEM hydrogel enriched with heart-specific ECM to compensate for the lack of cardiomyocyte (CM) maturation in the spheroid format. Instead of encapsulating pre-made 3D spheroids within the HEM hydrogel, we opted for different strategy, where the pre-gel solution of HEM is administered at the onset of 3D spheroid formation. In the former case, where pre-made 3D spheroids are encapsulated, the HEM envelops only the cells present on the surface of the spheroid. In contrast, in the latter case, the HEM fills the spaces between the cells and is distributed in the entire spheroid construct. Consequently, all the constituent cells forming the cardiac spheroid can directly engage with the heart-specific ECM, promoting cardiac maturation. Through a comparative analysis involving cardiac tissues crafted with HEM hydrogel, those formed without any hydrogel, and those produced using various commercially available hydrogels, we clearly demonstrated the maturation-boosting potential of HEM hydrogel (Fig. 2 and Supplementary Fig. 8a–b in the revised manuscript).

Second, we sought to address the issue of insufficient CM maturation by harnessing the dynamic flow generated by a microfluidic device. In contrast to conventional dynamic culture systems like spinner flasks or rotating bioreactors, our microfluidic chip offers a streamlined yet advanced approach to fluid dynamics. In particular, our chip harnesses interconnected microchannels between three culture chambers to efficiently deliver sophisticated medium flow to cardiac tissues, which is facilitated by a simple device such as an axial rocker (Fig. 3a and Supplementary Fig. 9a–b in the revised manuscript). We revealed that the dynamic flow in our microfluidic chip not only enhances the oxygen supply to the core of cardiac tissues, thereby improving their viability, but it also plays a pivotal role in cardiovascular development (Fig. 3d–e and Supplementary Fig. 10–12 in the revised manuscript). When we examined the sectioned cardiac tissues, we found that CMs were more prevalent at the border zone (the region of outer circle occupying over half the diameter) of our multicellular cardiac spheroid models due to the compartmentalization of three cell types. Upon exposure to dynamic flow, endothelial cells (ECs) spontaneously localized to the interior and cardiac fibroblasts (CFs) tended to be positioned in the exterior, a phenomenon previously reported in other studies (Reyat *et al.*, Front Cardiovasc Med. 2023;10:1156759,

Lee *et al.*, J Biol Eng. 2019;13:15). This suggests that our cardiac tissue model with dynamic flow could mimic the native heart development process, as validated in heart-forming organoids (Drakhlis *et al.*, Nat Biotechnol. 2021;39(6):737-746). Although many CMs were found at the border zone (the region of outer circle occupying over half the diameter) in our cardiac tissue, a mature thick CM layer was formed (Fig. 4i in the revised manuscript). This CM layer exhibited enhanced contractility and synchronized beating (Fig. 3g–j and Supplementary Movie 1–2 in the revised manuscript), making it useful for cardiotoxicity testing.

Additionally, we demonstrate that a combination of HEM hydrogel and microfluidic system can further improve cardiac maturation through a comprehensive analysis of transcriptomic profiles and identification of cardiac-specific microstructures (Fig. 4 and Supplementary Fig. 13–14 in the revised manuscript). In future, we consider integrating other physical stimuli (*e.g.*, cyclic stretch, electrical cue, *etc.*), known to be effective in promoting CM maturation (Ronaldson-Bouchard *et al.*, Nature. 2018;556(7700):239-243, Lu *et al.*, Theranostics. 2021;11(13):6138-6153), into our cardiac tissue engineering strategy. We have now provided the discussion above in the revised manuscript (page 28–29, line 664–681 and page 30, line 699–712).

Comment 1: The study is comprehensive, but it lacks a bit of understanding on the mechanism. For example, are HEM really contributing to myocardium regeneration? How much HEM are still present in the spheroids by the time the spheroids are implanted? Are all the benefits coming from having more healthy and matured cardiomyocytes in the spheroids? I think this is important to know considering the most novelty aspect of this paper is in the use of HEM.

Response 1: We sincerely appreciate the reviewer for these critical comments. Indeed, porcine decellularized heart-derived matrix alone, either in the form of an injectable hydrogel or patch, has shown therapeutic effects on myocardium regeneration, as evidenced by previous findings (Seif-Naraghi *et al.*, Sci Transl Med. 2013;5(173):173ra25, Shah *et al.*, ACS Appl Mater Interfaces. 2019;11(27):23893–23900). Despite such potential therapeutic effect of HEM, myocardium regeneration in our study is more likely attributed to the transplantation of healthy and matured cardiac cells. Prior to injecting the cardiac tissues into the infarcted myocardium, we induced the gentle fragmentation of cardiac tissues using collagenase for

reliable injection. Collagenase is an enzyme widely utilized for the degradation of decellularized tissue matrix and release of embedded cells (Boso *et al.*, *Biomedicines*. 2021;9(7):709, Quijano *et al.*, *Front Bioeng Biotechnol*. 2022;10:937239, Kim *et al.*, *Nat Commun*. 2022;13(1):1692). During this process, the majority of collagen components in the HEM are removed, leaving only trace amounts of other residual ECM constituents that could be implanted alongside the fragmented cardiac tissues. This suggests that the healing of the injured myocardium was mainly attributed to the cells in transplanted cardiac tissues rather than HEM and underscore the importance of the cellular condition of cardiac tissues in influencing the level of myocardium regeneration. In our study, HEM serves as a primary facilitator for enhancing the maturation of the constituent cells in cardiac tissues during *in vitro* culture, which was verified by comparative analysis with other hydrogels and RNA-sequencing analysis (Fig. 2 and 4 in the revised manuscript). In this respect, future studies need to focus on developing cardiac tissues that can fully harness the advantages of HEM and cardiac cells, aiming to establish more effective approaches for the healing of injured myocardium. We have now provided the discussion on this issue in the revised manuscript (page 26–27, line 620–637).

Comment 2: The significant improvement of cardiac function by the HEM-Cf group comparing to no HEM-Ps group is great. But it's unclear if this improvement is caused by HEM or perfusion on tissue spheroids. The improvement of perfusion on tissue spheroids is not a new finding and has been done before. In fact, I wonder if a cardiac spheroids without any ECM cultured in stirred tank bioreactor or a dish under constant rocking will provide the same level of regeneration.

Response 2: We greatly appreciate the reviewer for your critical comment. We agree on the reviewer's point. As the reviewer commented, we compared myocardium regeneration and cardiac function between the HEM-Cf and No HEM-Ps groups, but comparison of the No HEM-Cf and No HEM-Pf groups is missing in the originally submitted manuscript. To respond to the reviewer's comment, we have investigated the impact of dynamic flow in microfluidic chip and conventional well-plate on cardiac tissues under condition without HEM hydrogel. To this end, we have compared the No HEM-Pf group (cultured in well-type plates under constant rocking) with the No HEM-Cf group (cultured in microfluidic chips under constant rocking) in terms of mRNA and protein expressions in cardiac tissues

(Supplementary Fig. 21 in the revised manuscript). We found that the expression levels of CM, EC, and CF markers were significantly upregulated in the No HEM-Cf group compared with the No HEM-Pf group. These findings suggest that cardiac tissues fabricated in our microfluidic chip (No HEM-Cf) can exhibit a higher level of myocardium regeneration than those cultured in the well plate under constant rocking (No HEM-Pf), indicating superiority of our microfluidic system to conventional method for dynamic tissue culture. We have now provided these data in the Supplementary Figure 21 and provided the discussion above in the revised manuscript (page 25, line 592–599).

Comment 3: The histology sectioning of fibrotic regions should be quantified. This increase production of collagen is a major hallmark of fibrosis. This data should be quantified by experimental replicates.

Response 3: We sincerely appreciate the reviewer for this valuable comment. To address the reviewer's comment regarding quantification of the collagen deposition in our fibrosis model, we have now repeated the same experiments of fibrosis modeling but for higher concentration of TGF- β 1 (Fig. 6g in the revised manuscript). In the newly performed experiments, we increased the TGF- β 1 concentration from 10 ng/ml to 50 ng/ml to induce more aggressive fibrotic response. The new data, including bright-field images, immunofluorescent images, and contraction analysis, consistently demonstrate fibrotic phenotypes of TGF- β 1-treated cardiac tissues (Fig. 6h, i, k in the revised manuscript). Masson's trichrome (MT) staining images of the fibrotic cardiac tissues provided clear visualization of collagen deposition (Fig. 6j in the revised manuscript). To quantify the extent of fibrosis, we analyzed the collagen⁺ fibrosis area relative to the total area in cardiac tissue using ImageJ software. We observed a significant increase of the fibrosis area in cardiac tissues treated with 50 ng/ml TGF- β 1 when compared to normal cardiac tissues. These data suggest that the fibrotic cardiac tissues established in our culture platform recapitulate the pathophysiological features of cardiac fibrosis. We have now provided these data in the Figure 6g–k and stated the results in the revised manuscript (page 19, line 444–452).

Comment 4: To verify the model is effective, can an anti-fibrotic drug be used to show the reversal of the fibrosis progression?

Response 4: We thank the reviewer for this valuable comment. As per the reviewer's suggestion, we have conducted drug testing in our cardiac fibrosis model treated with 50 ng/ml TGF- β 1. We tested a well-known anti-fibrotic drug, losartan, which is an angiotensin II type 1 receptor antagonist known for its clinical efficacy in regressing myocardial fibrosis (Díez *et al.*, *Circulation*. 2002;105(21):2512-7, Raziyeva *et al.*, *Biomedicines*. 2022;10(9):2178). The anti-fibrotic effects of this drug were previously demonstrated in human cardiac fibrosis-on-a-chip model (Mastikhina *et al.*, *Biomaterials*. 2020;233:119741). In our study, the fibrotic tissues were treated with 5 μ M losartan on day 14 and cultured for additional 7 days (Fig. 6l in the revised manuscript). The disrupted CM structure in the fibrosis model was restored by the treatment of losartan (Fig. 6m–n in the revised manuscript). The impaired contraction profiles of the fibrosis model were also recovered by losartan treatment (Fig. 6o in the revised manuscript). These results suggest that cardiac tissues established in our culture platform can serve as a valuable cardiac fibrosis model for evaluation of anti-fibrotic efficacy of the drugs. We have now provided these data in the Figure 6l–o and stated the results in the revised manuscript (page 19–20, line 452–461).

Comment 5: It's unclear how many spheroids can fit inside the perfusion chip. For large animal studies in the future, large quantities of spheroids will be needed. How scalable is this platform?

Response 5: We thank the reviewer for this valuable comment. The microfluidic chip employed in our study featured three individual culture chambers dedicated to cardiac tissues within a single chip. One of the outstanding advantages of our chip lies in its axial rocker-based perfusion system, which obviates the necessity for additional tubing and syringe pumps. Thus, scaling up the system becomes remarkably straightforward – we merely stack the chips onto the rocker apparatus. As a single chip accommodating three cardiac tissues weighs approximately 20 grams, up to 250 chips could be housed and incubated simultaneously using an axial rocker with a permissible rocking weight of 5 kg. This scalability allows us to generate approximately 750 cardiac tissues per batch. Given that 10 cardiac tissues were transplanted per rat in our study, production in a scale-up batch would allow for animal study on about 75 rats. In addition, if the aim is to produce a greater quantity of cardiac tissues for large animal studies, it would be entirely feasible to fabricate the chips with adapted design that enables the cultivation of an increased amount of cardiac tissue by

expanding the number of chambers per chip. As already demonstrated in our prior research, we successfully developed perfusable microfluidic chips containing as many as 48 chambers, and it would be even feasible to design and manufacture the chips with up to 96 chambers (Jin *et al.*, *Adv Funct Mater.* 2018;28(37):1801954). In summary, our microfluidic chip system is exceptionally scalable with the chip design tailored to meet specific research objectives. We have now provided the discussion on this issue in the revised manuscript (page 27–28, line 647–653).

Comment 6: The Author clearly demonstrated a good level of maturation of these tissues, but it's unclear how the maturation level of these tissue spheroids compare with other engineered cardiac tissue that can induce cardiac alignment and actively maturing the tissues with electrical and/or mechanical stimulation. For regenerative medicine application, using decellularized matrix from porcine will always present some risks from residual DNA fragments if the matrix is not decellularized completely. Therefore, it's unclear how much benefit to tissue maturation does decellularized matrix provide when compared with maturation induced from electrical and/or mechanical stimulation and if this benefit justify the risk. This should be discussed.

Response 6: We thank the reviewer for this critical comment. Considering the safety concern regarding decellularized tissue-derived matrix, we conducted several experiments to ensure both *in vitro* and *in vivo* biocompatibility of our HEM (Supplementary Fig. 1 in the revised manuscript). We also confirmed that residual DNA content in our porcine HEM and native porcine heart was 35.6 ± 12.4 ng/mg and 4468.0 ± 97.1 ng/mg, respectively, which means that 99.2% of DNA was removed (Fig. 1b in the revised manuscript). Notably, previous studies have set the guideline for competent decellularization as residual DNA content not exceeding 50 ng/mg (Kasravi *et al.*, *Biomater Res.* 2023;27(1):10, Hussein *et al.*, *J Biomed Mater Res A.* 2018;106(7):2034-2047, Chakraborty *et al.*, *Biomater Sci.* 2020;8(5):1194-1215). Our HEM meets this criterion for clinical application. Actually, a wide array of decellularized matrix products have been used for clinical applications. For instance, a Phase 1 clinical trial assessed the utility of ECM hydrogel derived from porcine decellularized myocardium, named VentiGel (Ventrix, Inc.), in the treatment of myocardial infarction (Traverse *et al.*, *JACC Basic Transl Sci.* 2019;4(6):659-669). This study presented the safety and feasibility of decellularized heart matrix as a therapeutic option for patients with

myocardial infarction. Bovine decellularized vascular grafts including Artegraft® (Artegraft, Inc.) and ProCol® (LeMaitre Vascular Inc.) were also commercialized for clinical use (Pashneh-Tala *et al.*, Tissue Eng Part B Rev. 2016;22(1):68-100). There are commercially available products even if they contain more than 50 ng/mg of DNA contents, such as AlloPatch® (Aurora *et al.*, J Shoulder Elbow Surg. 2007;16(5 Suppl):S171-8).

Regarding the level of cardiac maturation, we need to compare the maturity of our cardiac tissues with that of the tissues engineered with other methods. As the reviewer commented, other engineering methods, such as cardiac alignment and electrical/mechanical stimulation, have been widely employed in advancing cardiac tissue maturation (Mao *et al.*, Nat Commun. 2023;14(1):2077, Ronaldson-Bouchard *et al.*, Nature. 2018;556(7700):239-243, Lu *et al.*, Theranostics. 2021;11(13):6138-6153). Notably, the utilization of decellularized tissue matrix in cardiac maturation provides distinct advantages over other methods, including a relatively short period for maturation and the ability to offer phenotype guidance even before cardiac specification (Mesquita *et al.*, Cells Tissues Organs. 2023;212(1):32-44, Tan *et al.*, Front Bioeng Biotechnol. 2022;10:831300). Throughout *in vivo* developmental processes, the combination of ECM interaction, non-myocyte interaction, and electrical/mechanical stimulation synergistically contributes to cardiac maturation (Scuderi *et al.*, Front Cell Dev Biol. 2017;5:50). This highlights the necessity for engineering approaches that integrate these elements concurrently into our cardiac tissues, resulting in synergistic impact of biochemical, electrical, and mechanical cues (Barbulescu *et al.*, Int J Mol Sci. 2022;23(21):13040). Since electrical/mechanical stimulation techniques are typically applied during the culture period, they can be readily integrated with HEM hydrogel culture for further promoting maturation of cardiac tissues. This combination approach would be worth being validated in the future studies. We have now provided the discussion on this issue in the revised manuscript (page 23–24, line 553–564 and page 30, line 699–712).

Comment 7: The platform doesn't allow the measurement of force of contraction (only demonstrated the amplitude of contraction) which is an important readout for cardiac drug testing. This should be discussed.

Response 7: We thank the reviewer for this valuable comment. As pointed out by the reviewer, the spherical shape of our cardiac tissues poses a limitation in directly measuring the force of contraction. Therefore, we used MUSCLEMOTION software to measure the

amplitude of contraction by optically assessing changes in the shape of the cardiac spheroids. MUSCLEMOTION software is extensively being utilized in 3D cardiac spheroid research, and the contraction amplitude derived from measurement of engineered heart tissues (EHT) with this software exhibited a strong linear correlation with force of contraction (Sala *et al.*, *Circ Res.* 2018;122(3):e5-e16). In addition, several studies have highlighted the effectiveness of using contraction amplitude for detecting drug-induced contractile responses (Beauchamp *et al.*, *Tissue Eng Part C Methods.* 2015;21(8):852-61) and contractile changes resulting from myocardial infarction (Richards *et al.*, *Nat Biomed Eng.* 2020;4(4):446-462). Therefore, when using our cardiac model to assess drug efficacy, we expect that measuring contraction amplitude allows us to infer drug-induced changes in contractile force readings, which will be sufficiently predictive of drug responsiveness. We have now provided the discussion on this issue in the revised manuscript (page 29, line 682–689).

Comment 8: The title of the paper needs some improvement. Microfluidic heart extracellular matrix doesn't make a lot of sense. I don't think the microfluidic component is so significant that it needs to be highlighted in the title. Using this type of microfluidic device to apply perfusion to spheroids is a very common approach. The focus should be on the heart extracellular matrix.

Response 8: Thank you for the valuable comment. As the reviewer suggested, we have revised the title of the paper to “Versatile human cardiac tissues engineered with perfusable heart extracellular microenvironment for biomedical applications”.

Again, the authors would like to thank the reviewer deeply for the time and effort in reviewing our manuscript.

RESPONSE TO REVIEWER #2

General comments: This is a well-designed experiment, in my opinion, there are still some flaws in this manuscript:

Response: We sincerely appreciate the reviewer for your insightful and meaningful comments on our manuscript. We have now tried to address all your comments, which has helped us to improve our manuscript substantially. We hope that all issues raised by the reviewer would be resolved with our responses. The revised parts in the manuscript have been highlighted in red.

Comment 1: The author chose three types of cardiovascular lineage cells (CMs, ECs, and CFs). Why were other cell types such as vascular smooth muscle cells not considered? How are the ratios and cell numbers of these three types of cells determined?

Response 1: We sincerely appreciate the reviewer for this valuable comment. In developing the cardiac tissue, we focused on gaining a balance between simplicity in fabrication and similarity to native heart tissue. Given that incorporation of excessive cell types complicates the manufacturing and culturing processes of cardiac tissues, we selected three cell types that effectively represent the major components of the heart – cardiomyocytes (CMs), endothelial cells (ECs), and cardiac fibroblasts (CFs). ECs and vascular smooth muscle cells (SMCs) are two major cell types of blood vessel, and as the reviewer pointed out, SMCs are crucial for maintaining vascular structural integrity and functionality. Nevertheless, we needed to constrain the number of cell types for reproducible fabrication of cardiac tissue. Consequently, we opted for ECs. This choice aligns with their recognized significance in cardiac development and regeneration (Talman *et al.*, *Front Cardiovasc Med.* 2018;5:101), as well as their frequent utilization in *in vitro* production of cardiac models (Campostrini *et al.*, *Circ. Res.* 2021;128:775–801). CFs were included in our cardiac tissues because they are recognized for promoting the maturation of CMs and playing a crucial role in regulating CM function through cellular crosstalk (Giacomelli *et al.*, *Cell Stem Cell.* 2020;26(6):862-879.e11). The mixing ratio of CM:EC:CF was determined as 2:1:1, based on several references reporting cardiac tissue production with these three cell types (Polonchuk *et al.*, *Sci Rep.* 2017;7(1):7005, Arai *et al.*, *PLoS One.* 2018;13(12):e0209162, Arai *et al.*, *Sci Rep.*

2020;10(1):8972). We have now added these statements above in the Results section of the revised manuscript (page 7, line 156–162) and clarified the rationale for the mixed cell ratio in the Methods section of the revised manuscript (page 33, line 777–779).

Comment 2: About the porcino's heart. How old is the donor porcine? Porcine HEM samples derived from ventricular or atrial tissue?

Response 2: Thank you for the valuable comment. Porcine HEM in our study was extracted from 6-month-old adult pigs, and left ventricles were used because they are the largest parts in the heart. The origin of porcine HEM has been described in detail in the Methods section of the revised manuscript (page 31, line 725–729).

Comment 3: The immune rejection of porcine HEM is very important for its possible subsequent clinical application. Digested ECM is also highly immunogenic. Is it reasonable for the author to discuss activated fibroblasts, leukocytes, M1 macrophages and M2 macrophages with only four molecules of α -SMA, CD11b, iNOS and CD206?

Response 3: We sincerely thank the reviewer for these constructive comments. Considering the safety concern regarding our decellularized tissue matrix, we conducted several experiments to assess both *in vitro* and *in vivo* biocompatibility of HEM (Supplementary Fig. 1 in the revised manuscript). To address the specific concern raised by the reviewer, we have now performed additional immunostaining for the markers of activated fibroblasts (Collagen type 1), leukocytes (CD45), M1 macrophages (CD80), and M2 macrophage (CD163). The quantification data from these additional immunostaining analyses showed that their expression patterns align with our previously presented data for α -SMA, CD11b, iNOS, and CD206. This consistency supports that a mild foreign body reaction occurred in response to the HEM hydrogel, but it returned to normal levels over time. We have now added these data in Supplementary Figure 1e–f and the statements above in the Results section of the revised manuscript (page 5, line 119–125).

In addition, we confirmed that our porcine HEM meets the established criterion for the safety of decellularized tissue-derived products. The residual DNA content in HEM and native porcine heart was 35.6 ± 12.4 ng/mg and 4468.0 ± 97.1 ng/mg, respectively, which means that 99.2% of DNA was removed (Fig. 1b in the revised manuscript). Notably,

previous studies have set the guideline for competent decellularization as residual DNA content not exceeding 50 ng/mg (Kasravi *et al.*, *Biomater Res.* 2023;27(1):10, Hussein *et al.*, *J Biomed Mater Res A.* 2018;106(7):2034-2047, Chakraborty *et al.*, *Biomater Sci.* 2020;8(5):1194-1215). Therefore, our porcine HEM meets this criterion for clinical application. Actually, a wide array of decellularized matrix products have been used for clinical applications. For instance, a Phase 1 clinical trial assessed the utility of ECM hydrogel derived from porcine decellularized myocardium, named VentriGel (Ventrix, Inc.), in the treatment of myocardial infarction (Traverse *et al.*, *JACC Basic Transl Sci.* 2019;4(6):659-669). This study presented the safety and feasibility of decellularized heart matrix as a therapeutic option for patients with myocardial infarction. Bovine decellularized vascular grafts including Artegraft® (Artegraft, Inc.) and ProCol® (LeMaitre Vascular Inc.) were also commercialized for clinical use (Pashneh-Tala *et al.*, *Tissue Eng Part B Rev.* 2016;22(1):68-100). There are commercially available products even if they contain more than 50 ng/mg of DNA contents, such as AlloPatch® (Aurora *et al.*, *J Shoulder Elbow Surg.* 2007;16(5 Suppl):S171-8). However, more comprehensive and in-depth analysis of decellularized tissue matrix in terms of safety and immunogenicity would be required in the future. We have now provided the discussion on this issue in the revised manuscript (page 23–24, line 553–564).

Comment 4: Can the simulated oxygen content truly reflect the oxygen content inside the heart tissue? Is there a way to directly or indirectly monitor oxygen levels? An example is the consumption rate of dissolved oxygen in the medium.

Response 4: We are grateful for this valuable comment. As suggested by the reviewer, we have newly investigated the indirect oxygen levels of cardiac tissues using the Image-iT™ red hypoxia reagent (ThermoFisher Scientific), as used in previous research on cardiac spheroids (Richards *et al.*, *Nat Biomed Eng.* 2020;4(4):446-462). This hypoxia reagent exhibits red fluorescence in response to decreased oxygen levels. We have compared three groups of cardiac tissues fabricated with HEM hydrogels under different flow conditions (HEM-Ps; Plate static, HEM-Pf; Plate flow, HEM-Cf; Chip flow) on day 14 of culture (Supplementary Fig. 10a in the revised manuscript). Fluorescence images of live cardiac tissues treated with the hypoxia reagent were taken using confocal microscopy at the intervals of the same thickness from the bottom of the cardiac tissues. A series of 15 z-slice confocal

images were captured at intervals of 10.03 μm and then stacked to generate a z-stack image with a total thickness of 140.42 μm . When quantifying the relative fluorescence intensity and fluorescence area (%) of cardiac tissues in each group, the HEM-Cf group showed significantly lower fluorescence signals. This outcome demonstrates that dynamic flow via our microfluidic chip can provide sufficient oxygen supply compared with no flow and other dynamic culture methods, leading to increased viability, functionality, and maturation of cardiac tissues (Fig. 3d–j in the revised manuscript). We have now provided these data in the Supplementary Figure 10a and stated the results in the revised manuscript (page 11, line 253–258). We have also described the detailed procedures in the Supplementary Methods (page 10, line 239–247)

Comment 5: The article shows that HEM shows improved efficacy in drug testing and disease modeling. Other methods should be used to prove the improvement of efficacy, such as analyzing the possible mechanism of improvement through transcriptomic or proteomic methods, and verifying the related molecules of improved efficacy through PCR and WB.

Response 5: We sincerely appreciate the insightful comment from the reviewer. In response to this comment, we have further analyzed our RNA sequencing data to elucidate underlying molecular mechanisms on the effects of HEM in improving drug testing and disease modeling (Supplementary Fig. 14 in the revised manuscript). We conducted Gene Ontology (GO) analysis with a focus on the differentially expressed genes (DEGs) upregulated in the HEM-Cf group compared to the No HEM-Cf group (Supplementary Fig. 14a in the revised manuscript). Notably, several GO terms associated with drug testing, including response to drug and response to cytokine, were significantly upregulated in the HEM-Cf group. Additionally, collagen-related GO terms, such as collagen binding, collagen trimer, collagen fibril organization, and collagen biosynthetic process, along with other GO terms related to ECM conferring elasticity and tensile strength, were enhanced by the incorporation of HEM hydrogel. This indicates that our HEM hydrogel can provide a suitable microenvironment for cardiac fibrosis modeling. Similarly, the upregulation of the GO term “arrhythmogenic right ventricular cardiomyopathy” suggests the applicability of our cardiac tissue for disease modeling associated with arrhythmias, such as long QT syndrome (LQTS).

Next, we examined the key DEGs expressed overlapping two or more GO terms

(Supplementary Fig. 14b in the revised manuscript). Among them, several genes such as *SPARC*, *SFRP2*, *GREM1*, *FOSL1*, *FBN1*, *FBLN5*, *COL1A1*, and *COL3A1* were significantly upregulated in the HEM-Cf group compared to the No HEM-Cf group, as well as in the HEM-Ps group compared to the No HEM-Ps group (Supplementary Fig. 14c in the revised manuscript). Secreted protein acidic and rich in cysteine (SPARC) is a collagen binding protein known to mediate collagen deposition and fibrosis in response to injury (Bradshaw *et al.*, *J Mol Cell Cardiol.* 2016;93:156-61, Yabluchanskiy *et al.*, *Curr Drug Targets.* 2013;14(3):276–286). The absence of this protein leads to defects in fibrillar collagen and impaired tissue healing process (Frangogianni *et al.*, *Physiol Rev.* 2012;92(2):635-88). Secreted frizzled-related protein 2 (SFRP2) acts as an antagonist of the Wnt signaling pathway, facilitating the fibrosis by increasing collagen deposition (Wu *et al.*, *Int J Biol Sci.* 2020;16(5):730–738, Feng *et al.*, *J Cardiovasc Dev Dis.* 2019;6(4):34). However, it also has a dual role in inhibiting cardiac fibrosis by regulating the BMP1 pathway (Wu *et al.*, *Int J Biol Sci.* 2020;16(5):730–738). Fibrillin-1 (FBN1) is one of the major glycoproteins conferring elasticity to cardiac tissue, and increased FBN1 deposition is accompanied by fibrosis progression (Bouzeghrane *et al.*, *Am J Physiol Heart Circ Physiol.* 2005;289(3):H982-91). The elevated expression of these genes in cardiac tissues fabricated with HEM hydrogel could support the deliberate cardiac fibrosis modeling.

Interestingly, SFRP2 treatment is known to promote the maturation of iPSC-derived CMs in terms of sarcomere structures, gap junctions, and electrophysiological properties (Hsueh *et al.*, *Sci Rep.* 2023;13:3920). *FBN1* gene is also considered as one of the markers for mature CM phenotypes (Xu *et al.*, *Stem Cells.* 2009;27(9):2163-74). Therefore, the upregulation of these genes in our cardiac tissues of the HEM-Cf group in comparison with No HEM-Cf group demonstrates the improved maturity attributed to HEM hydrogels. This may imply that the maturity of CMs enhanced in our cardiac tissues enable more precise response to drug. In addition, a previous study reported that overexpression of *Fos11* enhanced the cardiac functions in a mouse MI model (Wu *et al.*, *NPJ Regen Med.* 2021;6:36). Another study reported that treatment with Sfrp2 proteins in an MI model promoted resident cardiac c-Kit⁺ cells to express CM-specific genes, such as cardiac ion channels, leading to functional CM development (Hodgkinson *et al.*, *J Mol Cell Cardiol.* 2018;123:64–74). These findings suggest a potential mechanism behind the improved regenerative effects of our cardiac tissues in the HEM-Cf group. As per the reviewer’s suggestion, the upregulated expression of four genes (*SPARC*, *SFRP2*, *FOSL1*, and *FBN1*) was verified by qPCR analysis

(Supplementary Fig. 14d in the revised manuscript).

We have now provided these data in the Supplementary Figure 14 and stated the results and discussion in the revised manuscript (page 13–14, line 309–325 and page 26–27, line 621–627).

Comment 6: Given the potential immunogenicity of HEM, is it necessary to evaluate its therapeutic potential in a rat model of myocardial infarction? To assess the therapeutic potential of HEM, transplant rejection should be considered, for example by detecting molecules associated with transplant rejection.

Response 6: We are grateful for this critical comment. Indeed, there is substantial evidence from other studies supporting the potential therapeutic effect and safety of decellularized heart-derived matrix. Porcine decellularized heart matrix alone, whether in the form of an injectable hydrogel or a patch, has shown promising therapeutic effects on myocardium regeneration in a porcine or rat myocardial infarction model (Seif-Naraghi *et al.*, *Sci Transl Med.* 2013;5(173):173ra25, Shah *et al.*, *ACS Appl Mater Interfaces.* 2019;11(27):23893–23900). Moreover, a Phase 1 clinical trial assessed the utility of ECM hydrogel derived from decellularized porcine myocardium, named VentriGel (Ventrix, Inc.), in the treatment of MI (Traverse *et al.*, *JACC Basic Transl Sci.* 2019;4(6):659-669).

In our study, despite this potential therapeutic effect of HEM, myocardium regeneration is more likely attributed to the constituent cells in cardiac tissues fabricated using HEM and microfluidic devices. Prior to injecting the cardiac tissues into the infarcted myocardium, we induced the gentle fragmentation of cardiac tissues using collagenase for reliable injection. Collagenase is an enzyme widely utilized for the degradation of decellularized tissue matrix and release of embedded cells (Boso *et al.*, *Biomedicines.* 2021;9(7):709, Quijano *et al.*, *Front Bioeng Biotechnol.* 2022;10:937239, Kim *et al.*, *Nat Commun.* 2022;13(1):1692). During this process, the majority of collagen components in the HEM are removed, leaving only trace amounts of other residual ECM constituents that could be implanted alongside the fragmented cardiac tissues. In our study, HEM serves as a primary facilitator for enhancing the maturation of the constituent cells in cardiac tissues during *in vitro* culture (Fig. 2 and 4 in the revised manuscript). Overall, we speculate that the HEM would have limited direct effects on tissue regeneration and immune response *in vivo*. Nonetheless, as the reviewer suggested, in-depth future studies would be required to ensure

the immune tolerance of HEM by checking molecules associated with transplant rejection.

We have now provided the discussion on this issue in the revised manuscript (page 24, line 557–564 and page 26–27, line 620–637).

Comment 7: In Figures 2e and 2k, which part of the experiment did the statistical data of c-TnT come from?

Response 7: We apologize the reviewer for the confusion. The positive areas for cTnT in Figure 2e and 2k were quantified from immunofluorescent images stained along with CD31 shown in Figure 2d and 2j, respectively.

Comment 8: In Figure 3e and Figure 4f, the inner layer cells of the HEM-Pf and HEM-Ps groups partially express CD31, but the number of sensory cells is quite large, are they all endothelial cells?

Response 8: We are grateful for this critical comment. As the culture of our cardiac tissues progressed, we observed a tendency for CMs, ECs, and CFs to compartmentalize in the cardiac tissues. ECs spontaneously localized to the interior, and CFs tended to be positioned in the exterior, a phenomenon previously reported in other studies (Reyat *et al.*, Front Cardiovasc Med. 2023;10:1156759, Lee *et al.*, J Biol Eng. 2019;13:15). However, not all cells would be completely compartmentalized, as this phenomenon is not engineered and occurs spontaneously. Consequently, there could be uncompartimentalized CMs and CFs in the cardiac tissues. Additionally, CMs were primarily identified by their expression of cTnT in our study, and differentiated CMs from hiPSCs are generally known to contain more than 85% cTnT⁺ cells (Lian *et al.*, Proc Natl Acad Sci U S A. 2012;109(27):E1848-57, Mathur *et al.*, Sci Rep. 2015;5:8883). Despite our efforts to purify CMs using a glucose-free medium, there could be some remaining cTnT⁻ non-myocytes. This suggests that ECs, as well as other uncompartimentalized or unpurified cTnT⁻ cells, could be present within the inner layer of cardiac tissues. Furthermore, it is possible that some unhealthy cells were contained in the interior of the cardiac tissues in the HEM-Pf and HEM-Ps groups due to a lack of sufficient oxygen supply.

Comment 9: #1, #2 and #3 in Figure 4c are easily mistaken for 3 repetitions. suggested

modification.

Response 9: We are grateful for this thoughtful comment. As the reviewer suggested, we have now modified '#1, #2, and #3' in Figure 4c to '①, ②, and ③' to help the readers understand.

***Comment 10:* Figure 6a, the fabrication process of IPS in this schematic diagram is not the work of this manuscript**

Response 10: We are grateful for this comment. We have now modified the schematic diagram and description 'LQT patient' to 'LQT patient-derived iPSCs' in Figure 6a.

Again, the authors would like to thank the reviewer deeply for the time and effort in reviewing our manuscript.

RESPONSE TO REVIEWER #3

General comments: The manuscript by Min et al describes the development of spherical cardiac tissues by co-culture from iPSC derived cardiomyocytes and fibroblasts using a simple microfluidic device that provides gravity driven flow. The tissues are based on a hydrogel from decellularized porcine hearts. The work provides thorough characterization by a range of methods as well as a the use of the tissue for drug testing, disease modelling and in vivo therapy. Compared to other published methods, this system is easy to implement and could find use because of those reasons.

Response: We sincerely appreciate the reviewer for all the critical and insightful comments. We have tried to address all critical queries raised by the reviewer and to revise our manuscript thoroughly. We hope that the issues you raised would be resolved with our responses. The revised parts in the manuscript have been highlighted in red.

Comment 1: The tissues do not exhibit a well developed cardiac syncytium, from TnT staining. The tissues are rather immature compared to other methods.

Response 1: We thank the reviewer for this critical comment. To address this point, we have performed additional immunostaining of cardiac troponin T (cTnT) and obtained high-magnification images of the HEM-Cf group (Fig. 4f in the revised manuscript). Immunofluorescent images showed that individual cardiomyocytes (CMs) having different nuclei were structurally interconnected with well-developed sarcomeres. Furthermore, the synchronized Ca²⁺ transients observed in these cardiac tissues provided the evidence that cardiac tissues of the HEM-Cf group can function as a syncytium (Fig. 3j and Supplementary Movie 2 in the revised manuscript). These data suggest that cardiac tissues fabricated with HEM hydrogel and microfluidic chip successfully developed functional cardiac syncytium. We have now provided these data in the Figure 4f and stated the results in the revised manuscript (page 14, line 330–334).

Comment 2: Different levels of cell-gel compaction with control hydrogels vs decellularized ECM suggest there might be different mechanical properties at play that could contribute to the outcome.

Response 2: We appreciate the reviewer for this valuable comment. We agree with the reviewer's point that different mechanical properties of hydrogels may affect cell-gel compaction. In the HEM concentration test, 2 mg/ml HEM hydrogel with low elastic modulus enhanced differentiation of cardiac tissues compared to HEM hydrogels of high concentrations (4 and 6 mg/ml) with high elastic modulus (Fig. 1h–k in the revised manuscript). High level of cell-gel compaction can be facilitated by the hydrogel with low elastic modulus, leading to formation of more compact and condensed form of cardiac tissues. This contributes to augmentation of connectivity and interaction between the incorporated cells in cardiac tissues, which is crucial for cardiac differentiation (Le *et al.*, *In Vitro Cell Dev Biol Anim.* 2018;54(7):513-522). Therefore, difference in mechanical properties between ECM hydrogels and other commercially available hydrogels could affect differentiation and maturation of cardiac tissues. To compare the cell-gel compaction of the hydrogels, we have now measured the mechanical properties of HEM hydrogels, commercially available hydrogels, and other tissue-derived ECM hydrogels (Supplementary Fig. 8c in the revised manuscript). We found that the larger the elastic modulus of the hydrogel, the lower cell-gel compaction, leading to generation of the cardiac tissue with the larger size (Fig. 2c, i in the revised manuscript). In particular, the two types of Matrigel groups did not induce the formation of compact spherical tissues and showed poor cellular connections due to the scattered distribution of incorporated cells (Fig. 2b–d in the revised manuscript). On the other hand, HEM hydrogel exhibited the best performance for cardiac tissue formation in comparison with other hydrogel groups showing similar levels of elastic modulus and cell-gel compaction, such as fibrin gel, stomach-derived ECM hydrogel, and muscle-derived ECM hydrogel (Fig. 2 and Supplementary Fig. 8c in the revised manuscript). These data suggest that beneficial effects of HEM hydrogel could be attributed to both high level of cell-gel compaction and heart-specific ECM components. We have now added this data in Supplementary Figure 8c and stated the results in the revised manuscript (page 9–10, line 216–231). We have also described the detailed procedures for measuring the viscoelastic properties of hydrogels in the Supplementary Methods (page 2, line 55–59)

Comment 3: Comparison of decellularized ECM from different organs is elegant.

Response 3: We thank the reviewer for this considerate comment. In our research, we

conducted a comparative analysis between HEM hydrogel and ECM hydrogels sourced from intestine, stomach, and muscle tissues. This investigation was undertaken to ascertain the impacts of the heart-specific ECM components present in HEM hydrogel. We appreciate the reviewer's thoughtful comment again.

Comment 4: For ventricular myocardium, quiescent tissues are better, ie a sign of improved maturation. High spontaneous beating of ventricular tissues is not necessarily a plus.

Response 4: We appreciate the reviewer for this critical comment. As pointed out by the reviewer, adult ventricular CMs are electrically quiescent in the absence of external stimulation. In contrast, immature CMs, such as fetal CMs, exhibit spontaneous beating. Similarly, hiPSC-derived CMs are known to beat spontaneously due to their resemblance to fetal properties (Karbassi *et al.*, Nat Rev Cardiol. 2020;17(6):341–359). The human fetal heart typically beats between 120–160 beats per minute (BPM), while gradually decreasing to 60–100 BPM during maturation to the adult heart (Steinburg *et al.*, PeerJ. 2013;1:e82). In our cardiac tissues of the HEM-Cf group, the BPM falls within the range of 60–80, which is close to the level of normal adult heart beats. This suggests that although our hiPSC-derived cardiac tissues may not reach the level of maturity seen in the adult heart, the CMs of our cardiac tissues exhibit a noteworthy advancement in the maturity, compared with typical hiPSC-derived CMs that are considered highly immature. We have now provided the discussion on this issue in the revised manuscript (page 29, line 689–698).

Comment 5: Sarcomeres are not well developed compared to other approaches.

Response 5: We thank the reviewer for this valuable comment. To address this point, we have additionally conducted immunostaining of α -actinin and comparative analysis of sarcomere structures in cardiac tissues developed under four different conditions: No HEM-Ps, HEM-Ps, No HEM-Cf, and HEM-Cf (Fig. 4e in the revised manuscript). Notably, the cardiac tissues in the HEM-Cf group exhibited the most aligned and longest sarcomere structures when compared to the other groups. This observation suggests that the combination of HEM hydrogel and dynamic flow in microfluidic chip substantially promotes CM maturation. Specifically, the average sarcomere length in the HEM-Cf group measured $1.87 \pm 0.14 \mu\text{m}$, a

level similar to that seen in late-stage CMs and other models like engineered heart tissues (EHT) (Lundy *et al.*, Stem Cells Dev. 2013;22(14):1991-2002, Mannhardt *et al.*, Stem Cell Reports. 2020;15(4):983-998). However, this measured value still falls short of the sarcomere length seen in fully matured CMs (Ronaldson-Bouchard *et al.*, Nature. 2018;556(7700):239-243). This indicates that further modification, such as provision of additional cues (*e.g.*, electrical signal, alignment, *etc.*), may be necessary to promoting their maturation to the fullest extent. The application of such combination approach needs to be validated in the future studies. We have now provided these data in the Figure 4e and stated the results and discussion in the revised manuscript (page 14, line 326–330 and page 30, line 699–712).

Comment 6: More convincing evidence is needed for the presence of T-tubules.

Response 6: We appreciate the reviewer for this valuable comment. To respond to this comment, we performed wheat germ agglutinin (WGA) staining again to detect T-tubules, and we obtained high-magnification images from the HEM-Cf group (Fig. 4f in the revised manuscript). These images confirmed the development of T-tubules in the α -actinin⁺ sarcomere structures, particularly evident in the HEM-Cf group on day 14 of the culture. Furthermore, scanning electron microscopy at a higher magnification revealed the organization of T-tubules alongside the Z-lines within the sarcomere structures (Fig. 4l in the revised manuscript) (Yang *et al.*, Circ Res. 2014;114(3):511-23, Ergir *et al.*, Sci Rep. 2022;12(1):17409). Collectively, these findings provide the evidence that T-tubules were developed in the cardiac tissues fabricated with HEM hydrogel and dynamic flow in our microfluidic chip. We have now provided these data in the Figure 4f and stated the results in the revised manuscript (page 14, line 334–335 and page 15, line 356–357).

Comment 7: The hallmark of LongQT syndrome is prolonged action potential. It should be shown if one is to claim disease modelling of LongQT. Additionally, all the same contraction duration with Long QT is shown. One would expect prolongation, but it may not be necessary, so long Aps are prolonged.

Response 7: We greatly appreciate the reviewer for the critical comment. When conducting contraction analysis in Long QT syndrome (LQTS) modeling, we applied an electrical stimulus of 1 Hz to address the differences in beats per minute (BPM) between each cardiac

tissue. The contraction duration was equalized owing to electrical pacing, while the relaxation time was lengthened in LQT2 cardiac tissue model. In response to the reviewer's suggestion, we also assessed the electrophysiological characteristics of LQTS cardiac tissue models using microelectrode array (MEA) (Fig. 6e–f in the revised manuscript). The cardiac tissues fabricated with LQT2 patient iPSC-derived cardiomyocytes (LQT2-CMs) and LQT2 patient iPSC-derived cardiac fibroblasts (LQT2-CFs) exhibited significantly prolonged field potential duration (FPD) when compared with normal cardiac tissues. Moreover, cardiac tissues fabricated with either LQT2-CMs and normal CFs or normal CMs and LQT2-CFs also showed prolonged FPD compared to the normal cardiac tissues. However, their FPD prolongations were shorter than those of cardiac tissues fabricated using both LQT2-CMs and LQT2-CFs. These results suggest that while either LQT2-CMs or LQT2-CFs can serve individually to depict LQTS traits, their collaborative utilization in constructing a cardiac tissue produces a synergistic effect, resulting in a more precise representation of LQTS characteristics. We have now provided these electrophysiological data in the Figure 6e–f and stated the data in the revised manuscript (page 18–19, line 429–438). We have also described the detailed procedures in the Supplementary Methods (page 16, line 398–402)

Comment 8: Mason's Trichrome evidence of fibrosis is not convincing. Collagen is appears very clearly blue in Mason's trichrome. More convincing evidence of fibrosis is needed.

Response 8: We thank the reviewer for the valuable comment. To address the reviewer's comment regarding the collagen deposition in our fibrosis model, we have now conducted the same experiments of fibrosis modeling but for higher concentration of TGF- β 1 (Fig. 6g in the revised manuscript). In the newly performed experiments, we increased the TGF- β 1 concentration from 10 ng/ml to 50 ng/ml to induce more aggressive fibrotic response. The new data, including bright-field images, immunofluorescent images, and contraction analysis, consistently demonstrate fibrotic phenotypes of TGF- β 1-treated cardiac tissues (Fig. 6h, i, k in the revised manuscript). Masson's trichrome (MT) staining images of the fibrotic cardiac tissues provided clear visualization of collagen deposition (Fig. 6j in the revised manuscript). To quantify the extent of fibrosis, we analyzed the collagen⁺ fibrosis area relative to the total area in cardiac tissue using ImageJ software. We observed a significant increase of the fibrosis area in cardiac tissues treated with 50 ng/ml TGF- β 1 when compared to normal

cardiac tissues. These data suggest that the fibrotic cardiac tissues established in our culture platform recapitulate the pathophysiological features of cardiac fibrosis. We have now provided these data in the Figure 6g–k and stated the results in the revised manuscript (page 19, line 444–452).

Comment 9: Strong functional data on transplantation are strong in terms that both echo and hemodynamics measurements were provided. But the experiment ends at 4 weeks, it should be taken to 6 weeks, that is the time when pathological remodeling is over after MI. Often treatments may fail at 6 weeks.

Response 9: We appreciate your valuable comment on this point. We agree with your opinion that some cell therapies may prove ineffective 6 weeks post-myocardial infarction (MI). To address this concern, we have conducted a follow-up experiment to assess whether the improvement in cardiac function achieved through transplantation of cardiac tissues of the HEM-Cf group would remain consistent for up to 6 weeks (Fig. 7b–e in the revised manuscript). The left ventricular ejection fraction (LVEF) and fractional shortening (LVFS) were significantly higher in the HEM-Cf group than in the other groups (No treatment and No HEM-Ps) at 1 week, and this trend continued for up to 6 weeks. Furthermore, the left ventricular internal diameters at end-diastolic (LVIDd) and end-systolic (LVIDs) phases were significantly smaller in the HEM-Cf group at 6 weeks. The septal wall thickness (SWT) was thicker in the HEM-Cf group compared to the other groups. The results from 6-week post-transplantation study demonstrated a significant restoration of cardiac function in the HEM-Cf group, which showed a similar trend to our prior data conducted up to 4 weeks. Importantly, this improvement persisted even at the 6-week time point, affirming the durability of the positive outcomes. We have now provided these data in the Figure 7b–e and stated the results in the revised manuscript (page 20–21, line 475–485).

Comment 10: Details and dosing of tissue injection are needed.

Response 10: We apologize the reviewer for the confusion. We previously described methodological details of cardiac tissue transplantation in the manuscript. In more detail, the rats were re-anesthetized for transplantation using isoflurane inhalation, intubated, and mechanically ventilated. The chests of the animals were reopened, and the infarct area with

collagen deposition was inspected. Subsequently, we injected cardiac tissues at two spots in the border zone of infarcted myocardium. The cardiac tissues, harvested 9 days after fabrication, were fragmented with collagenase type 2 for reliable injection. Ten fragmented cardiac tissues containing 1.0×10^6 RFP-CMs were suspended in 50 μ l of PBS for injection into one rat. The injection dose of cardiac tissues was standardized based on the quantity of CMs, which was identical to the number of CMs transplanted in the previous study (Park *et al.*, Nat Commun. 2019;10(1):3123). The No treatment group was injected with PBS only.

We have now added detailed information about the method of cardiac tissue transplantation in the revised manuscript (page 36, line 840–847).

Comment 11: The initial injury was mild, often people use larger infarcts, so that Ejection fraction goes down to 30%. Mild injury is easier to reverse.

Response 11: We thank the reviewer for this critical comment. The animal model utilized in our study was an ischemic/reperfusion injury model designed to mimic clinical situations. We performed additional experiments to explore the change of ejection fraction in MI control group up to 6 weeks post-MI (Fig. 7c in the revised manuscript). Our observation revealed that, even after reperfusion, ejection fraction gradually declined over time. Simultaneously, there was a progressive thinning of the wall thickness, leading to the deterioration of the heart and adverse remodeling (Fig. 7e in the revised manuscript). Eventually, the ejection fraction reached approximately 33% at 6 weeks in the MI control group.

Comment 12: Provide measurements for ventricular wall thickness.

Response 12: We thank the reviewer for this comment. We have quantified and included the data on the septal wall thickness (SWT) and posterior wall thickness (PWT) (Fig. 7e in the revised manuscript). Notably, SWT showed a significant increase in the HEM-Cf group starting from 1-week post-transplantation, and this increase continued up to 6 weeks. In contrast, the PWT showed no significant difference during the follow-up period, as it was not subjected to ischemic/reperfusion injury (Park *et al.*, Exp Mol Med. 2021;53(9):1423-1436). We have now provided these data in the Figure 7e and stated the results in the revised manuscript (page 20–21, line 482–485).

Comment 13: Evidence of integration is weak, there is no optical mapping or any other evidence of functional integration of the graft.

Response 13: We sincerely appreciate the reviewer for bring this critical point to our attention. In cell therapies with CMs for cardiac regeneration, one of the critical concerns is the potential occurrence of arrhythmias resulting from poor integration of the transplanted cells into the host myocardium (Almeida *et al.*, Card Electrophysiol Clin. 2015;7(2):357-70). To provide the evidence of functional integration of CMs in the infarcted hearts, we have performed electrocardiogram (ECG) measurements at the endpoint using a Langendorff isolated heart perfusion system (Supplementary Fig. 19 in the revised manuscript). ECG data were collected from the *ex vivo* isolated hearts for 10 minutes and analyzed with a real-time heartbeat classification algorithm, generating waterfall plots for all detected heartbeat. In comparison to the No injury group (sham operation) and the HEM-Cf group, the No treatment group and the No HEM-Ps group showed typical pathological Q-waves and ST-segment elevation, the indicative of MI. Heart rhythm analysis of ECG can provide the numbers of events including premature beats, ventricular tachycardia, fibrillation, asystole, and cardiac pause. The HEM-Cf group showed a lower incidence of premature beats without fatal ventricular arrhythmia when compared to the No treatment group and No HEM-Ps group. Moreover, the PR, RR, QRS, and corrected QT (QTc) intervals recorded in the HEM-Cf group resembled those of the No injury group. Additionally, the HEM-Cf group showed relatively higher engraftment of transplanted CMs compared to the No HEM-Ps group, and the expression of connexin-43 (CX43) was well-organized and displayed connectivity with neighbor CMs up to 6 weeks (Fig. 8d and Supplementary Fig. 20 in the revised manuscript). These electrophysiological and structural data suggest that the transplanted CMs successfully achieved functional integration within the host myocardium, mitigating the risk of arrhythmias. We have now provided these data in the Supplementary Figure 19–20 and stated the results in the revised manuscript (page 22–23, line 523–538). We have also described the detailed procedures for ECG measurement in the Supplementary Methods (page 18, line 439–448)

Comment 14: Please check the text in the methods on endothelial cell differentiation. Where do ECs in the tissues come from?

Response 14: We thank the reviewer for valuable comment. To fabricate human cardiac tissues in our study, human umbilical vein endothelial cells (HUVECs) were incorporated with hiPSC-derived CMs and hiPSC-derived CFs, as described in the Methods section. They consistently expressed the EC marker (CD31) in the fabricated cardiac tissues. We also screened optimal co-culture medium by comparing CD31 expression to induce their stable growth in fabricated cardiac tissues (Supplementary Fig. 7 in the revised manuscript). In future studies, it would be necessary to produce cardiac tissues using hiPSC-derived ECs with the same origin as CMs and CFs.

Comment 15: Validate assumptions for the use of Brinkman equation for tissue modelling. If flow is assumed through the hydrogel/tissue of course modelling will give the higher levels of oxygen. However, compact and dense tissues such as the heart tissue in vivo do not have flow (convection) in the tissue space. Flow happens through the capillary bed. The use of this equation then assumes cell density is inherently low, which is opposite from the main claim in this paper, that this is a dense cardiac tissue.

Response 15: We thank the reviewer for these critical and valuable comments. As the reviewer commented, *in vivo* heart receives flow through the capillary bed, not in the tissue space. However, for cultivating 3D cell structures like spheroids and organoids in a healthy and sustained manner, external fluid flow is currently required. To address this need, various microfluidic chip designs have been invented and employed for culturing spheroids and organoids (Fang *et al.*, Adv Funct Mater. 2023;33(19):2215043, Jun *et al.*, Sci Adv. 2019;5(11):eaax4520, Homan *et al.*, Nat Methods. 2019;16(3):255-262). In our study, we utilized microfluidic chips customized for our cardiac tissues and conducted a simulation analysis to assess their impact on oxygen concentration within cardiac tissues. In response to the reviewer's suggestion, we have now re-performed computational simulation of oxygen concentration in cardiac tissues with and without dynamic flow in our microfluidic chips (Fig. 3b–c in the revised manuscript). Cardiac tissues were assumed to be dense cell spheroids, instead of the hydrogel with the Brinkman's equation. Diffusion coefficients of the oxygen in the medium and spheroid were set to $2.4 \times 10^{-5} \text{ cm}^2/\text{s}$ and $3.0 \times 10^{-6} \text{ cm}^2/\text{s}$ respectively, according to previous studies (Jin *et al.*, Adv Funct Mater. 2018;28(37):1801954, Richards *et al.*, Nat Biomed Eng. 2020;4(4):446-462). This assumption leads to a steeper drop in oxygen concentration with increasing depth into the

spheroids, compared to the simulation using the previous assumption. Notably, the dynamic culture maintained higher oxygen concentrations in cardiac tissues than the static culture, which is consistent with our prior results. Consequently, we confirmed that dynamic flow in our microfluidic chips enhances the overall oxygen concentrations in cardiac tissues, suggesting the augmentation of the viability of cells inside the cardiac tissues. We have now revised the data in Figure 3b–c and described the simulation parameters in the Supplementary Methods (page 9–10, line 235–238).

Comment 16: There could be some flow through the voids of the tissue due to the contraction (that would almost act like a pumping motion), but this all has to be avlaidated. Brinkman equation is for porous media, is this really porous media if cells are there and connected to one another as the authors claim.

Response 16: We appreciate the reviewer for this thoughtful comment. The simulation of oxygen concentration in our study has been used to infer the effects of dynamic flow with microfluidic chips. As we responded to the comment #15, new computational simulation has been conducted assuming dense cell spheroids instead of the hydrogel with Brinkman's equation. Diffusion coefficients of the oxygen in the medium and spheroid were set to $2.4 \times 10^{-5} \text{ cm}^2/\text{s}$ and $3.0 \times 10^{-6} \text{ cm}^2/\text{s}$ respectively, according to previous studies (Jin *et al.*, *Adv Funct Mater.* 2018;28(37):1801954, Richards *et al.*, *Nat Biomed Eng.* 2020;4(4):446-462). Then, we confirmed that dynamic flow enhanced the oxygen concentration in our cardiac tissues. As the reviewer highlighted, it would be a highly valuable project to simulate cardiac tissue containing the voids in tissue whose volume changes with cardiac contraction. In our current study, any form of voids in the cardiac tissues has not yet been detected, which needs to be improved (Fig. 3e in the revised manuscript). This might be improved through the development of cardiac tissues containing endocardial chambers through the incorporation of endocardial cells (Lewis-Israeli *et al.*, *Nat Commun.* 2021;12(1):5142). In future work, it is necessary to fabricate cardiac organoids with void chambers and investigate the fluid flow and oxygen diffusion in such more realistic heart model. We have now added this discussion in the revised manuscript (page 30, line 712–717).

Comment 17: Along those lines, please provide cell density across the diameter of the tissue, as well as viable or apoptotic cell density to prove the claim cell density was

uniform.

Response 17: We appreciate the reviewer for your valuable comment. One of the limitations of our simulation results is that they are based on a relatively short period (~60 mins), mainly serving to infer that the dynamic flow provided by microfluidic chips improves the oxygen supply to the inner cells of cardiac tissues. Since our primary focus was on the early stages of culture, we performed the sectioning of cardiac tissues one day after fabrication to measure cell density (DAPI⁺ cells/mm²) in center zones and border zones of cardiac tissues (Supplementary Fig. 9c in the revised manuscript). We found that the overall cell density was consistent and uniform, regardless of the location within the construct. This suggests that our computational simulation, which assumed to be dense cell spheroids as the reviewer suggested, was conducted properly for comparing the initial oxygen supply. Additionally, we have provided the data on cell density and apoptotic cells (%) in the center zones and border zones of the HEM-Ps group, HEM-Pf group, and HEM-Cf group 14 days after the fabrication (Supplementary Fig. 10b–c in the revised manuscript). We found that the cell density of cardiac tissues in all groups was maintained uniformly across the whole construct. The expression of Caspase 3-positive apoptotic cells in HEM-Ps and HEM-Pf group was significantly elevated inside compared with outside of cardiac tissues. In contrast, the HEM-Cf group exhibited a relatively lower expression of apoptotic cells. This alignment with our prediction using simulation analysis supports the conclusion that dynamic flow via microfluidic chips enhances the viability of cells within our cardiac tissues. We have now added the data in the Supplementary Figure 9c, 10b–c and stated the results in the revised manuscript (page 10–11, line 244–246 and page 11, line 260–263).

***Comment 18:* Critical details are missing in the methods about how exactly the tissue was formed, about the MI model and details about the tissue injection into the heart are also missing.**

Response 18: We apologize the reviewer for the confusion. We previously described several important methods in the manuscript, including cardiac tissue generation, transplantation, and MI model induction. To fabricate human cardiac tissues, CMs, CFs, and ECs were mixed at a 2:1:1 ratio. Then, mixed cells were encapsulated at 2.0×10^5 cells per 20 μ l of HEM hydrogel. The formed cardiac tissues were transferred to microfluidic chips and cultured on

an axial rocker on the second day. For transplantation, cardiac tissues were harvested 9 days after fabrication and fragmented with collagenase type 2 for reliable injection. Then, ten fragmented cardiac tissues containing 1.0×10^6 RFP-CMs were suspended in 50 μ l of PBS for injection into one rat. The injection dose of cardiac tissues was standardized based on the quantity of CMs, which is identical to the number of CMs transplanted in the previous study (Park *et al.*, Nat Commun. 2019;10(1):3123).

To induce myocardial ischemia, the left anterior descending (LAD) artery was ligated for 1 hour using a 7-0 prolene suture. After 1 hour of ischemia induction, the knot was removed to allow for reperfusion of the LAD. The chest was aseptically closed and disinfected after surgery. One week after ischemia/reperfusion injury, base echocardiography was performed prior to transplantation of cardiac tissues. After that, the rats were re-anesthetized for transplantation using isoflurane inhalation, intubated, and mechanically ventilated. The chests of the animals were reopened, and the infarct area with collagen deposition was inspected. Subsequently, we injected the fragmented cardiac tissues at two spots in the border zone of infarcted myocardium. The No treatment group was injected with PBS only. We have added detailed information about the method of cardiac tissue transplantation in the revised manuscript (page 36, line 840–847).

Comment 19: It would be really nice to show movies of contraction and Ca²⁺ transients.

Response 19: We appreciate the reviewer for your valuable comment. We have now provided the movies showing contraction and Ca²⁺ transients of cardiac tissue in the HEM-Cf group on day 14 of the culture (Supplementary Movie 1 and 2 in the revised manuscript). The cardiac tissue in the movies was the sample of HEM-Cf group displayed in Figure 3j.

Again, the authors would like to thank the reviewer deeply for the time and effort in reviewing our manuscript.

REVIEWERS' COMMENTS

Reviewer #1 (Remarks to the Author):

The Authors have addressed all my comments. In my opinion this paper can be published in the journal.

Reviewer #2 (Remarks to the Author):

It is a pleasure to receive revised manuscript and comments from the author. I found that the author added a large number of experiments based on some of my suggestions, such as investigating the indirect oxygen levels of cardiac tissues, in-depth evaluation of the safety of HEM, re-analyzing RNA sequencing data, etc. The author also provides a detailed supplementary description of the issues that I have doubts about, such as immunity and transplantation risks. Overall, this study constructed engineered functional cardiac tissue spheroids from a variety of cells and successfully applied them to drug testing and disease simulation, which has great application prospects. I have no problem anymore.

Reviewer #3 (Remarks to the Author):

The authors answered most of my questions well. I am still not convinced about the T-tubules. The cells are immature and the tissues are not that big, there is really no incentive for T-tubule development. The added WGA staining does not appear conclusive. It could also be non-specific. The manuscript would not be hurt much if the claim about T-tubules was removed.

RESPONSE TO REVIEWER #1

Comment: The Authors have addressed all my comments. In my opinion this paper can be published in the journal.

Response: We sincerely thank the reviewer for your thoughtful comment on our manuscript. Your critical and valuable feedback has greatly improved the manuscript. We appreciate the effort you put into reviewing our manuscript.

RESPONSE TO REVIEWER #2

Comment: It is a pleasure to receive revised manuscript and comments from the author. I found that the author added a large number of experiments based on some of my suggestions, such as investigating the indirect oxygen levels of cardiac tissues, in-depth evaluation of the safety of HEM, re-analyzing RNA sequencing data, etc. The author also provides a detailed supplementary description of the issues that I have doubts about, such as immunity and transplantation risks. Overall, this study constructed engineered functional cardiac tissue spheroids from a variety of cells and successfully applied them to drug testing and disease simulation, which has great application prospects. I have no problem anymore.

Response: We sincerely appreciate the reviewer for your thoughtful comment on our manuscript. Owing to your critical and valuable feedback, our manuscript has been greatly improved. Thank you for the time and effort taken to review our manuscript.

RESPONSE TO REVIEWER #3

Comment: The authors answered most of my questions well. I am still not convinced about the T-tubules. The cells are immature and the tissues are not that big, there is really no incentive for T-tubule development. The added WGA staining does not appear conclusive. It could also be non-specific. The manuscript would not be hurt much if the claim about T-tubules was removed.

Response: As the reviewer suggested, we have removed the data and claim about the presence of T-tubules in our manuscript (Fig. 4f and 4l in the revised manuscript). Owing to your critical and valuable suggestions, our manuscript has been greatly improved. Thank you for the time and effort taken to review our manuscript.